# Imp/IGF2BP levels modulate individual neural stem cell growth and division through *myc* mRNA stability

Tamsin J Samuels[1], Aino I Järvelin[1], David Ish-Horowicz[1,2], Ilan Davis[1]*

[1]Department of Biochemistry, The University of Oxford, Oxford, United Kingdom; [2]MRC Laboratory for Molecular Cell Biology, University College, London, United Kingdom

**Abstract** The numerous neurons and glia that form the brain originate from tightly controlled growth and division of neural stem cells, regulated systemically by important known stem cell-extrinsic signals. However, the cell-intrinsic mechanisms that control the distinctive proliferation rates of individual neural stem cells are unknown. Here, we show that the size and division rates of *Drosophila* neural stem cells (neuroblasts) are controlled by the highly conserved RNA binding protein Imp (IGF2BP), via one of its top binding targets in the brain, *myc* mRNA. We show that Imp stabilises *myc* mRNA leading to increased Myc protein levels, larger neuroblasts, and faster division rates. Declining Imp levels throughout development limit *myc* mRNA stability to restrain neuroblast growth and division, and heterogeneous Imp expression correlates with *myc* mRNA stability between individual neuroblasts in the brain. We propose that Imp-dependent regulation of *myc* mRNA stability fine-tunes individual neural stem cell proliferation rates.

*For correspondence:
ilan.davis@bioch.ox.ac.uk

**Competing interests:** The authors declare that no competing interests exist.

## Introduction

The many cells of the brain are produced through the highly regulated repeated divisions of a small number of neural stem cells (NSCs). NSCs grow and divide rapidly in order to supply the cells of the developing brain, but must be restrained to prevent tumour formation. Individual NSCs produce characteristic lineages of progeny cells (*Kriegstein and Alvarez-Buylla, 2009*; *Merkle et al., 2007*), which vary in number suggesting differences in division and growth rates during development. However, the mechanisms differentially regulating the growth and division of individual NSCs are currently unknown.

Many of the processes and factors regulating neurogenesis are conserved between mammals and insects, making *Drosophila* an excellent model system to study NSC regulation (*Homem and Knoblich, 2012*). During *Drosophila* neurogenesis, NSCs, also known as neuroblasts (NBs), divide asymmetrically, budding-off a small progeny cell, the ganglion mother cell (GMC), which divides into neurons that progress through differentiation. During larval neurogenesis, the NB divides on average once every 80 min (*Homem et al., 2013*) and regrows between divisions to replace its lost volume, maintaining the proliferative potential of the cell (*Homem and Knoblich, 2012*). However, average measurements of growth and division mask considerable heterogeneity between the behaviour of individual NBs in the brain over developmental time. Individual NBs produce unique lineages of neurons (*Pereanu and Hartenstein, 2006*), with characteristically different clone sizes (*Yu et al., 2013*). Individual NBs also have differing division frequencies (*Hailstone et al., 2019*) and terminate division at different times (NB decommissioning) (*Yang et al., 2017a*). This individual control ensures that the appropriate number of each neuron type is produced in the correct location during the construction of the brain. Systemic signals such as insulin and ecdysone signalling drive NB growth and division, with a particularly strong influence at the transitions between developmental stages (*Chell and*

**eLife digest** The brain is a highly complex organ made up of huge numbers of different cell types that connect up to form a precise network. All these different cell types are generated from the repeated division of a relatively small pool of cells called neural stem cells. The division of these cells needs to be carefully regulated so that the correct number and type of nerve cells are produced at the right time and place. But it remains unclear how the division rate of individual neural stem cells is controlled during development.

Controlling these divisions requires the activity of countless genes to be tightly regulated over space and time. When a gene is active, it is copied via a process called transcription into a single-stranded molecule known as messenger RNA (or mRNA for short). This molecule provides the instructions needed to build the protein encoded within the gene.

Proteins are the functional building blocks of all cells. The conventional way of controlling protein levels is to vary the number of mRNA molecules made by transcription. Now, Samuels et al. reveal a second mechanism of determining protein levels in the brain, through regulating the stability of mRNA after it is transcribed.

Samuels et al. discovered that a key regulatory protein called Imp controls the growth and division of individual neural stem cells in the brains of developing fruit flies. The experiments showed that Imp binds to mRNA molecules that contain the code for a protein called Myc, which is known to drive cell growth and division in many different cell types. Both human Imp and Myc have been implicated in cancer.

Using a technique that images single molecules of mRNA, Samuels et al. showed that the Imp protein in stem cells stabilises the mRNA molecule coding for Myc. This means that when more Imp is present, more Myc protein gets produced. Thus, the level of Imp in each individual neural stem cell fine-tunes the rate at which the cell grows and divides: the higher the level of Imp, the larger the stem cell and the faster it divides.

These findings underscore how important post-transcriptional processes are for regulating gene activity in the developing brain. The methods used in this study to study mRNA molecules in single cells also provide new insights that could not be derived from the average measurements of many cells. Similar methods could also be applied to other developmental systems in the future.

Brand, 2010; Géminard et al., 2009; Homem et al., 2014; Ren et al., 2017; Rulifson et al., 2002; Sousa-Nunes et al., 2011; Syed et al., 2017). However, the reproducible heterogeneity between individual NBs implies the existence of an unknown local or cell-intrinsic signal, acting in addition to the systemic signals to determine the proliferation of each NB.

The temporal regulation of NB proliferation and progeny fate has been well studied in the embryo and larva, and many key factors have been identified (Doe, 2017; Li et al., 2013; Miyares and Lee, 2019; Rossi et al., 2017). The developmental progression of larval NBs is characterised by the levels of two conserved RNA-binding proteins (RBPs), IGF2 mRNA-binding protein (Imp/IGF2BP2) and Syncrip (Syp/hnRNPQ) (Liu et al., 2015). Imp and Syp negatively regulate each other and are expressed in opposing temporal gradients through larval brain development (Liu et al., 2015): Imp level in the NB declines through larval development while Syp level correspondingly increases. Imp and Syp play numerous key roles in larval neurogenesis. The levels of Imp and Syp are known to determine the different types of neuron produced by the NBs over time, through post-transcriptional regulation of the transcription factor (TF) *chinmo* (Liu et al., 2015; Ren et al., 2017). The loss of Syp results in an enlarged central brain, in part due to an increase in NB proliferation rate (Hailstone et al., 2019). In pupal NBs, declining Imp expression allows NB shrinkage and Syp promotes NB termination (Yang et al., 2017a). Temporal regulation of the Imp/Syp gradients depends on the upstream temporal patterning system (Narbonne-Reveau et al., 2016; Ren et al., 2017; Syed et al., 2017). The timing and rates of change of these RBP levels differ substantially between classes of NB, and to a lesser degree between NBs of the same class (Liu et al., 2015; Syed et al., 2017; Yang et al., 2017a). However, it is unknown if the intrinsic levels of Imp and Syp in each NB play a role in controlling the growth and division rates of individual NBs during their main proliferative window in the larva.

Imp and Syp are RBPs and can modify the protein complement of a cell via post-transcriptional modulation of mRNA localisation, stability and translation rates (*Boylan et al., 2008*; *Geng and Macdonald, 2006*; *Hobor et al., 2018*; *McDermott et al., 2012*; *McDermott et al., 2014*; *Medioni et al., 2014*; *Munro et al., 2006*). Cell growth and proliferation are classically thought to be regulated at the level of transcription by pro-proliferative TFs. Various signalling pathways converge to promote cell growth and proliferation through transcriptional upregulation of the conserved TF and proto-oncogene, Myc (*Dang, 2012*; *Delanoue et al., 2010*; *Levens, 2010*; *Teleman et al., 2008*). Myc interacts with a binding partner, Max, to exert widespread transcriptional effects, binding upwards of 2000 genes in *Drosophila* (*Orian et al., 2003*). In *Drosophila,* Myc is best known for its role in promoting cell growth through increased ribosome biogenesis (*Grewal et al., 2005*), and also accelerates progression through the G1 phase of the cell cycle in the developing wing, though this does not affect overall cell cycle length (*Johnston et al., 1999*). It is unclear whether the transcriptional activation of pro-proliferative TFs, such as Myc and its downstream targets, is overlaid by post-transcriptional regulatory mechanisms executed by RBPs, such as Imp and Syp, which could increase the precision and flexibility of the system.

Here, we examine the role of the Imp/Syp temporal gradient in regulating NB size and division during larval neurogenesis. We show that the upregulation of Imp increases NB division and size, while Syp influences these processes indirectly via its negative regulation of Imp. We use a genome-wide approach to determine the mRNA targets bound by Imp in the brain and identify *myc* mRNA among the top 15 targets of Imp. Single molecule fluorescent in situ hybridisation (smFISH) shows that *myc* mRNA is stabilised by Imp, leading to increased Myc protein levels, NB growth and proliferation. We compare NB types with different Imp levels and find that low Imp levels result in unstable *myc* mRNA, which restrains NB growth and division. Finally, at an earlier time point, when Imp expression is heterogeneous between individual NBs, we find that higher Imp correlates with increased *myc* mRNA half-life. We propose a model in which Imp post-transcriptionally regulates *myc* mRNA stability to fine-tune individual NB size and division rate in their appropriate developmental context.

## Results

### Imp promotes type I NB growth and division

To investigate the roles of the opposing Imp and Syp gradients in the NB, we used RNAi knockdown to manipulate the level of these RBPs (*Figure 1—figure supplement 1*). We studied the type I NBs, the most numerous NB type in the brain, which are also very convenient to analyse, as they have a simple division hierarchy with each asymmetric division producing a GMC that divides only once more to produce two neurons or glia (*Bello et al., 2008*; *Boone and Doe, 2008*; *Bowman et al., 2008*). In the wandering L3 stage (wL3) brains all type I NBs express high levels of Syp and low of Imp (*Figure 1—figure supplement 1A*). We depleted Syp or Imp from the NBs with *Syp* knockdown and *Imp* knockdown RNAi constructs using the GAL4-UAS system, driven by *insc-GAL4* (*Betschinger et al., 2006*). In NBs Imp and Syp negatively regulate each other and therefore the *Syp* knockdown results in Imp upregulation (*Figure 1—figure supplement 1B*) (*Liu et al., 2015*). We distinguished between direct effects of Syp depletion and indirect effects due to upregulated Imp expression by analysing *Imp Syp* double knockdown mutants (*Figure 1—figure supplement 1C*) (*Yang et al., 2017a*). We also examined Imp overexpression brains, but the UAS overexpression construct only produces a very limited upregulation of Imp in the type I NB at the wL3 stage (*Figure 1—figure supplement 1D*), as previously observed (*Liu et al., 2015*; *Yang et al., 2017a*). Therefore we primarily use the *Syp* knockdown to upregulate Imp.

We first examined the roles that Imp and Syp play in influencing type I NB size. Our results show that higher Imp promotes larger size of type I NBs at wL3, and Syp acts indirectly through its negative regulation of Imp. Imp-depleted NBs are almost half the size of *wild type* NBs and NBs that overexpress Imp are 1.4-fold larger in midpoint area (*Figure 1A,A′*, Materials and methods). Syp-depleted NBs are 1.5-fold larger than *wild type*. We tested whether this effect is direct or indirect by studying the size of NBs in the *Imp Syp* double knockdown. Our results show that Imp depletion suppresses the increase in NB size observed in *Syp* knockdown mutants, which indicates that Syp only plays an indirect role in type I NB size, through its repression of Imp.

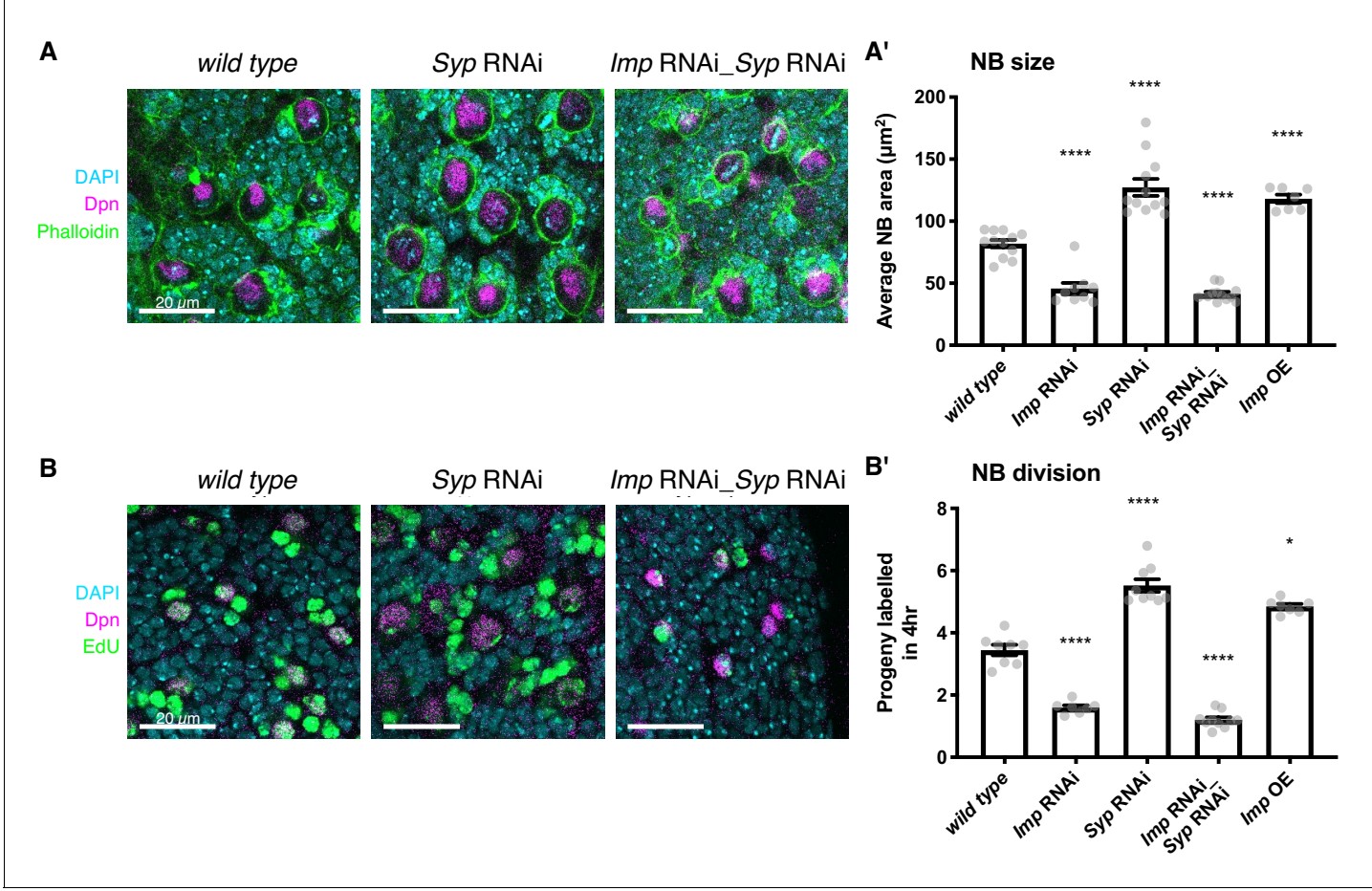

**Figure 1.** Elevated Imp levels increase NB proliferation and size. (**A**) Phalloidin was used to stain F-actin, marking the perimeter of each type I NB in the central brain (the largest cells, identified with Deadpan (Dpn) immunofluorescence (IF)). The area of each NB was measured at its largest point, and the average NB size per brain is plotted in (**A'**). NBs with diffuse Dpn (indicating nuclear envelope breakdown during mitosis) were excluded. (**B**) Larval brains were cultivated ex vivo with 25 µM EdU for four hours. All cells that underwent DNA synthesis in S phase are labelled with EdU. Dpn IF labels type I NBs. The number of progeny produced by each NB in the central brain was compared in *wild type*, *Imp* RNAi, *Syp* RNAi, double *Imp Syp* RNAi and Imp overexpression (OE) brains. The average number of progeny per NB in each brain is plotted in (**B'**). In **A'** and **B'**, significance was calculated using a one-way ANOVA and Dunnett's multiple comparisons test, with comparison to *wild type*. **p<0.01, ***p<0.001, ****p<0.0001. Each grey point represents one wL3 brain and for each genotype at least seven brains were measured, from three experimental replicates.

The online version of this article includes the following figure supplement(s) for figure 1:

**Figure supplement 1.** *Syp* RNAi and double *Imp Syp* RNAi distinguishes the roles of Imp and Syp.

NB size is affected by both cell growth and division rate so we then tested whether NB division rate is also sensitive to Imp levels. We incubated ex vivo explanted brains in 5-ethynyl-2'-deoxyuridine (EdU)-containing media for four hours to label the progeny cells produced during this time (see Materials and methods). The number of labelled progeny was decreased by more than half in the *Imp* RNAi brains compared to *wild type* (*Figure 1B,B'*), which suggests that the decreased NB size in the *Imp* knockdown is not due to an increased division rate. The number of progeny was increased 1.4-fold in the Imp overexpressing brains and increased 1.6-fold in the *Syp* RNAi brains, in which Imp is strongly upregulated, compared to *wild type.* This phenotype is consistent with the increased proliferation rate previously observed in *Syp* knockdown brains with ex vivo culture and live imaging (*Hailstone et al., 2019*). However, the increased proliferation was lost in the *Imp Syp* double knockdown brains. These results, together with our previous findings that Imp overexpression prevents NB shrinkage in the pupa and extends NB lifespan (*Yang et al., 2017a*), suggest that low levels of Imp in the late larval NBs restrains NB growth and division, ensuring the brain growth is limited appropriately during its development.

## Imp binds hundreds of mRNA targets in the brain, including *myc*

Imp is an RBP, so is likely to exert its function in the NB through regulation of the RNA metabolism of its key target mRNA transcripts. In an effort to identify strong candidate targets, we identified the transcripts bound by Imp in the brain. To achieve this aim we performed Imp RNA immunoprecipitation and sequencing (RIPseq) in larval brain lysates (see Materials and methods). We identified 318 mRNA targets that were significantly enriched in the Imp pulldown compared to input brain RNAseq (using the thresholds DESeq2.padj < 0.01 and DESeq2.log2FoldChange > 2) (*Figure 2—figure supplement 1A,B*, *Supplementary file 1*). The list of targets includes known Imp targets such as *chickadee* (target rank: 37) (*Medioni et al., 2014*), as well as mRNAs that have previously been shown to be regulated by Imp. Imp binds *syp* mRNA (target rank: 103), which indicates a post-transcriptional mechanism for the previously observed negative regulation of Syp by Imp (*Liu et al., 2015*). Another Imp target is *chinmo* (target rank: 55), which is known to be post-transcriptionally regulated by Imp to determine the progeny fate of NBs in the mushroom body (MB), the centre for memory and learning. Chinmo is also regulated by Imp in type II NBs (*Liu et al., 2015*; *Ren et al., 2017*; *Syed et al., 2017*) and during NB self renewal (*Dillard et al., 2018*; *Narbonne-Reveau et al., 2016*). Imp binds a number of long non-coding RNAs, including *CR43283/cherub* (target rank: 5). *cherub* is also a binding target of Syp and facilitates Syp asymmetric segregation during type II NB division (*Landskron et al., 2018*). The large number of Imp targets identified by RIPseq indicates that Imp has a broad range of roles in the developing brain. Imp has been shown to regulate mRNA localisation, stability, and translation (*Degrauwe et al., 2016*). Our results suggest that examining the Imp targets will provide further insight into the role of Imp in neurogenesis and the critical importance of post-transcriptional regulation.

To identify the key candidate mRNA targets responsible for the Imp NB size and division phenotypes, we examined the gene ontology (GO) annotations of the top 40 Imp targets (*Figure 2A*). We searched for genes annotated to play a role in cell growth, cell size, cell cycle and neural development, as well as regulatory genes with RNA-binding or DNA-binding function (*Figure 2B*, *Supplementary file 1*). We identified *myc* (target rank: 13) as the top candidate that could explain the Imp phenotype, based on these GO categories. As discussed in the introduction, *myc* is a master transcription factor regulator of growth and division in diverse model systems. In *Drosophila* it is primarily known as a driver of cell growth (*Grewal et al., 2005*), and is a determinant of self renewal in the type II NB (*Betschinger et al., 2006*). We also identified a second member of the Myc transcriptional network, *mnt*, as an mRNA target bound by Imp (target rank: 36). Mnt competes with Myc for binding to Max, and promotes opposed transcriptional effects (*Loo et al., 2005*; *Orian et al., 2003*). We first focussed on *myc,* and later investigated *mnt*. *myc* is the 13th most enriched target of Imp and is a very promising candidate as a direct mediator of the Imp phenotype in NBs.

To further examine the interaction between Imp and *myc* mRNA, we reanalysed a previously published dataset of Imp iCLIP (individual nucleotide resolution cross-linking and immunoprecipitation) performed in S2 cells (*Hansen et al., 2015*). The iCLIP data shows that Imp directly binds the *myc* transcript (*Figure 2—figure supplement 1C*), which supports our identification of *myc* mRNA as an Imp target in the brain. The iCLIP experiment identifies Imp binding sites primarily in the *myc* untranslated regions (UTRs) and binding signal is enriched in the extended 3′ UTR of the longer mRNA isoform. In our brain Imp RIPseq dataset, we also see reads throughout the extended 3′ UTR, suggesting that Imp binds to the long *myc* mRNA isoform (*Figure 2—figure supplement 1D*). Notably, the full *myc* 3′ UTR extension is expressed in the brain (*Figure 2—figure supplement 1E*) but it is truncated early in the S2 cells (*Figure 2—figure supplement 1F*), so the fully extended transcript in the brain may contain additional Imp binding sites. The results in S2 cells support our identification of *myc* mRNA as a target of Imp in the brain, highlighting the hypothesis that Imp is a key regulator of *myc* in the NB.

## Myc expression is regulated by Imp levels

To test the hypothesis that Myc protein levels are regulated by Imp, we used antibody staining in *wild type* and knockdown type I NB lineages. We found that Imp is required to maintain correct Myc levels in the NB. We observed Myc protein expression in type I NBs, but not in the surrounding GMCs or neurons (*Figure 3A*). Myc protein level was increased more than 2-fold in the *Syp* RNAi NBs compared to *wild type* (*Figure 3B*, quantitated in 3C), while this effect was lost in the double

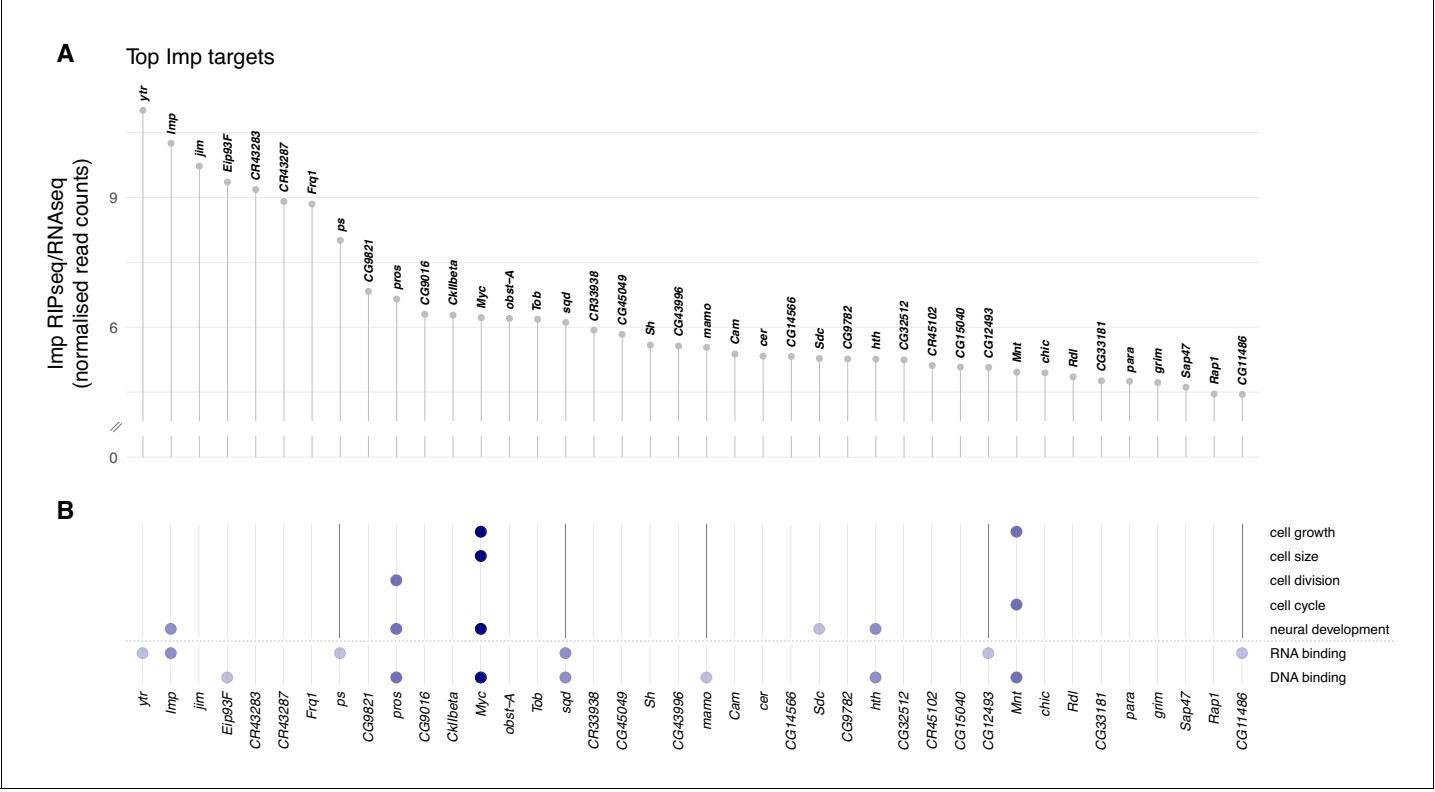

**Figure 2.** Imp RNA targets in the *D. melanogaster* wL3 brain. (**A**) Ranked top 40 Imp RIPseq targets relative to baseline RNA expression as measured by RNAseq. Non-coding RNAs that overlap other genes are excluded. (**B**) Genes in panel A mapped to gene ontology (GO) terms related to cellular growth and division, neural development, and regulatory functions RNA- and DNA-binding. Each dot indicates the gene is annotated to one or more GO terms in that category. The colour of the dots reflects the total number of GO categories each gene maps to, out of the seven investigated. The online version of this article includes the following figure supplement(s) for figure 2:

**Figure supplement 1.** Imp RIPseq identifies mRNA targets of Imp in the brain.

*Imp Syp* depleted NBs. Directly overexpressing Imp resulted in a small increase in Myc protein level (1.2-fold increase on *wild type* level) (**Figure 3C**). The effect of Imp overexpression on Myc protein level is smaller than that in *Syp* knockdown NBs as the overexpression construct produces a smaller upregulation of Imp (**Figure 1—figure supplement 1**). *Imp* knockdown produced a small decrease in Myc protein level (**Figure 3C**), as expected because Imp levels are already very low in *wild type* type I NBs. These data indicate that Imp upregulation increases Myc protein level in the NB, while Syp's effect on Myc is indirect, as it requires Imp.

We next examined the effect of Imp and Syp on Mnt, the antagonist of Myc, also identified as an Imp target. Using antibody staining, we found that Mnt protein is expressed in the type I NB, as well as in the progeny cells of the lineage (**Figure 3—figure supplement 1A**). However, knockdowns of *Imp* and *Syp* have no effect on the levels of Mnt protein. Therefore, we conclude that Mnt is not likely to be a key target responsible for the NB growth and division phenotype of Imp.

We then asked whether the upregulation of Myc by Imp could be responsible for the phenotype of increased type I NB growth and division. We overexpressed the Myc open reading frame (ORF) in type I NBs (**Figure 3—figure supplement 1B**, Materials and methods) and found a significant 1.3-fold increase in NB size (**Figure 3D**). *Myc* knockdown produced a small and not significant decrease in NB size. We used a *Myc Syp* double knockdown to confirm that upregulated Myc is responsible for the increased size of *Syp* knockdown NBs (in which Imp is upregulated). We found that the increased NB size in the *Syp* knockdown is lost in the *Myc Syp* double knockdown brains (*Myc_Syp* RNAi NBs are 0.7x the size of *wild type*), supporting the hypothesis that Imp regulates NB size through upregulation of Myc.

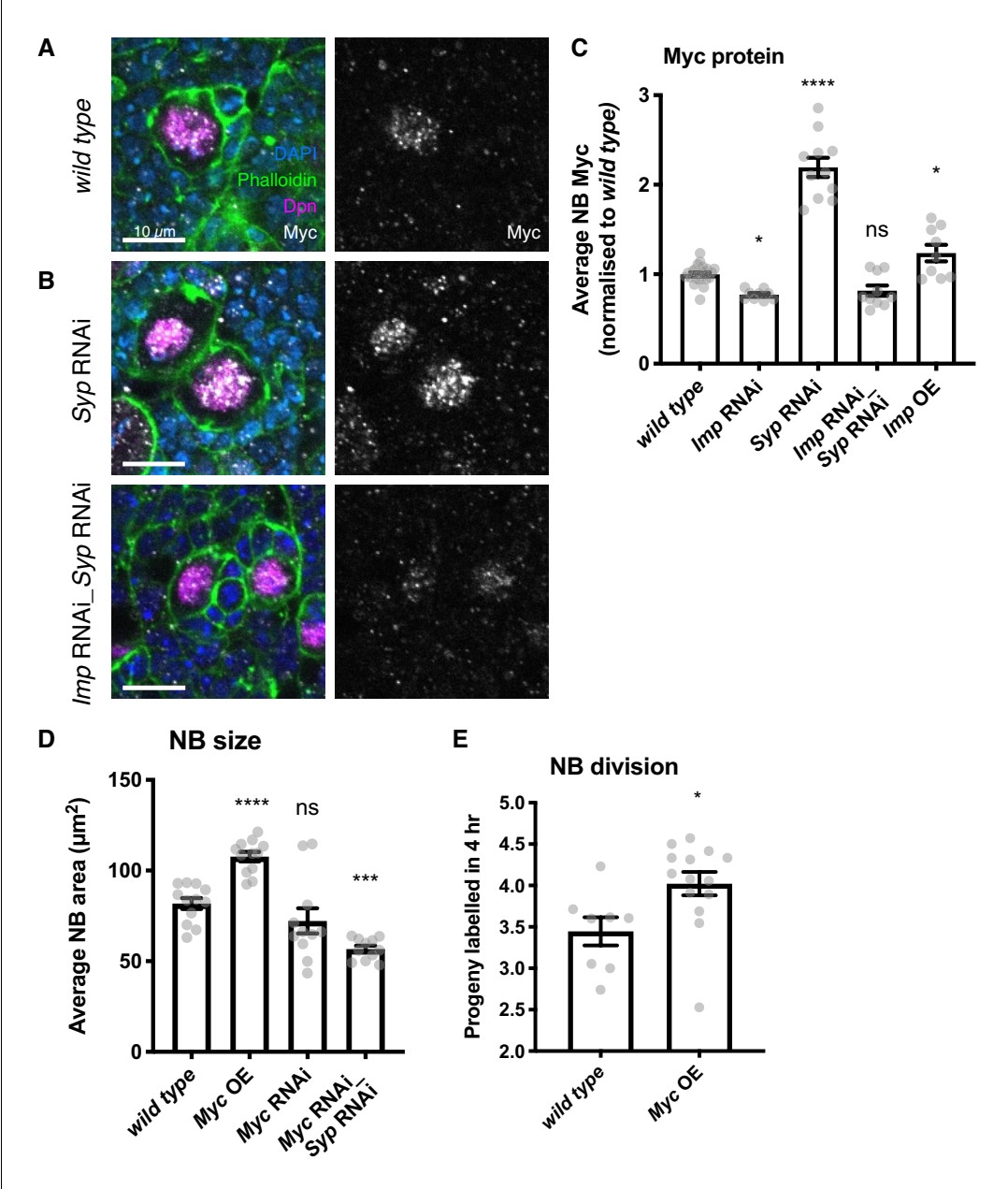

**Figure 3.** Imp upregulates Myc protein expression, which in turn determines NB division rate and size. (A) Antibody staining against Myc protein, with NBs labelled with Dpn. Myc protein is restricted to the NB in the *wild type* type I lineage. (B) In the *Syp* knockdown, Myc protein is increased in the NB, but this increase is lost in the *Imp Syp* double knockdown. The average Myc IF signal in NBs per brain is quantitated in C. D) Myc overexpression increases NB size, measured as NB area at the widest point. *Myc* RNAi results in a non-significant decrease in NB size. *Myc Syp* double knockdown reverses the phenotype of *Syp* single knockdown, resulting in small NBs compared to *wild type*. (E) EdU staining to count progeny produced in a 4 hr incubation shows that overexpression of Myc increases NB proliferation. Significance was calculated using a one-way ANOVA and Dunnett's multiple comparisons test, with comparison to *wild type*. ns non significant, *p<0.05, ***p<0.001, ****p<0.0001 Each grey point represents one wL3 brain and for each genotype at least eight brains were measured, from three experimental replicates.

The online version of this article includes the following figure supplement(s) for figure 3:

**Figure supplement 1.** *mnt* and *myc* are targets of Imp.

We tested the effect of Myc overexpression on type I NB division rate, and observed an increased division rate in the Myc OE compared to *wild type* (*Myc OE*: 4.04 EdU-labelled progeny per NB, *Figure 3E*). The observed increase in division rate is a surprising result as previous work in the wing disc showed that Myc overexpression increased cell size without affecting division rate (*Johnston et al., 1999*), highlighting that Myc could regulate cell size and division rate in distinct ways in different tissue contexts. In the NB, we find that increased Myc protein levels can explain the increased size and division rate that occur in response to overexpressing Imp. However, Imp levels are very low in wL3 *wild type* type I NBs (*Figure 1—figure supplement 1*), which may limit Myc protein expression and restrain NB growth and division.

## Imp stabilises *myc* mRNA

In order to further characterise the regulation of *myc* mRNA by Imp, we visualised *myc* mRNA transcripts using smFISH in type I NBs (*Yang et al., 2017b*). The two annotated RNA isoforms of *myc* are identical except that the longer isoform includes a 3' UTR extension of 5.7 kb (*Figure 4A*) (FlyBase, *Thurmond et al., 2019*). This additional UTR sequence potentially includes substantial regulatory sequence, including multiple binding sites for Imp according to iCLIP in S2 cells (*Hansen et al., 2015*) (*Figure 2—figure supplement 1C*), which could allow differential regulation of the two isoforms. smFISH probes against the *myc* intron and common exon show *myc* transcription and mature *myc* transcripts in the type I NB (*Figure 4A,B*, *Figure 4—figure supplement 1A*, *Supplementary file 2*). Co-staining with the common exon probe and a long-UTR-specific probe, showed that all cytoplasmic transcripts in the type I NB are positive for both probes (*Figure 4A,C*). This result shows that the extended UTR isoform of *myc* (*myc^long^*) is the predominant isoform expressed in the NB. Therefore, we used probes specifically against the *myc^long^* isoform for the following quantitative experiments.

Imp binds to *myc* mRNA and could upregulate Myc protein either through increasing *myc* mRNA levels or increasing Myc translation. To distinguish between these possibilities, we stained brains with *myc^long^*-specific smFISH probes and quantitated the RNA expression in individual NBs within the mixed-cell tissue (*Figure 4—figure supplement 1B,C*, Materials and methods, *Mueller et al., 2013*). We measured the effects of *Imp* knockdown, Imp upregulation using the *Syp* knockdown, and suppression in the *Imp Syp* double knockdown. Due to the minimal upregulation of Imp with the Imp overexpression construct (*Figure 1—figure supplement 1*) and correspondingly small upregulation of Myc protein (*Figure 3C*), we did not quantitate the *myc* mRNA expression in the Imp overexpression brains (*Figure 4—figure supplement 1B*). The number of *myc^long^* transcripts per NB is significantly reduced in the *Imp* knockdown, and is significantly increased in the *Syp* knockdown (*Figure 4D,E*). The transcript number is similar to *wild type* levels in the *Imp Syp* double knockdown, showing that Imp, rather than Syp, is the primary regulator of the number of *myc^long^* transcripts observed in the NB. We interpret our results as showing that the increase in *myc* transcript number observed when Imp is upregulated causes the observed increase in Myc protein level. In contrast, Imp is unlikely to upregulate Myc protein levels primarily through an increase in *myc* translation efficiency, although the data does not exclude the possibility that this mechanism makes a minor contribution to Myc protein upregulation.

The number of mature transcripts is affected by both transcription rate and mRNA stability. In order to distinguish between a role for Imp in regulating *myc* transcription rate or *myc* transcript stability, we used smFISH measurements to estimate the transcription rate and mRNA half-life of *myc^long^* in each NB (*Bahar Halpern and Itzkovitz, 2016*). We used the average intensity of a single transcript to calculate the number of nascent transcripts at the transcription foci, which indicates the relative transcription rate (*Mueller et al., 2013*, Materials and methods). We found that while the number of nascent transcripts is not significantly changed in the *Imp* knockdown or the *Syp* knockdown, it is significantly reduced in the *Imp Syp* double knockdown (*Figure 4F*). We used this measurement to estimate the transcription rate and showed that *myc^long^* transcription is unchanged in the single knockdowns, but is significantly reduced in the *Imp Syp* double knockdown (*Figure 4G*, Materials and methods, [*Bahar Halpern and Itzkovitz, 2016*]). This change in *myc* transcription in *Imp Syp* double knockdown NBs is unexpected, and may be an indirect effect through other transcription factors that Imp and Syp regulate, or a feedback loop of Myc autoregulation.

To determine the post-transcriptional role of Imp in regulating *myc* transcript level we calculated the *myc* mRNA half-life, allowing direct comparison between genotypes despite differing

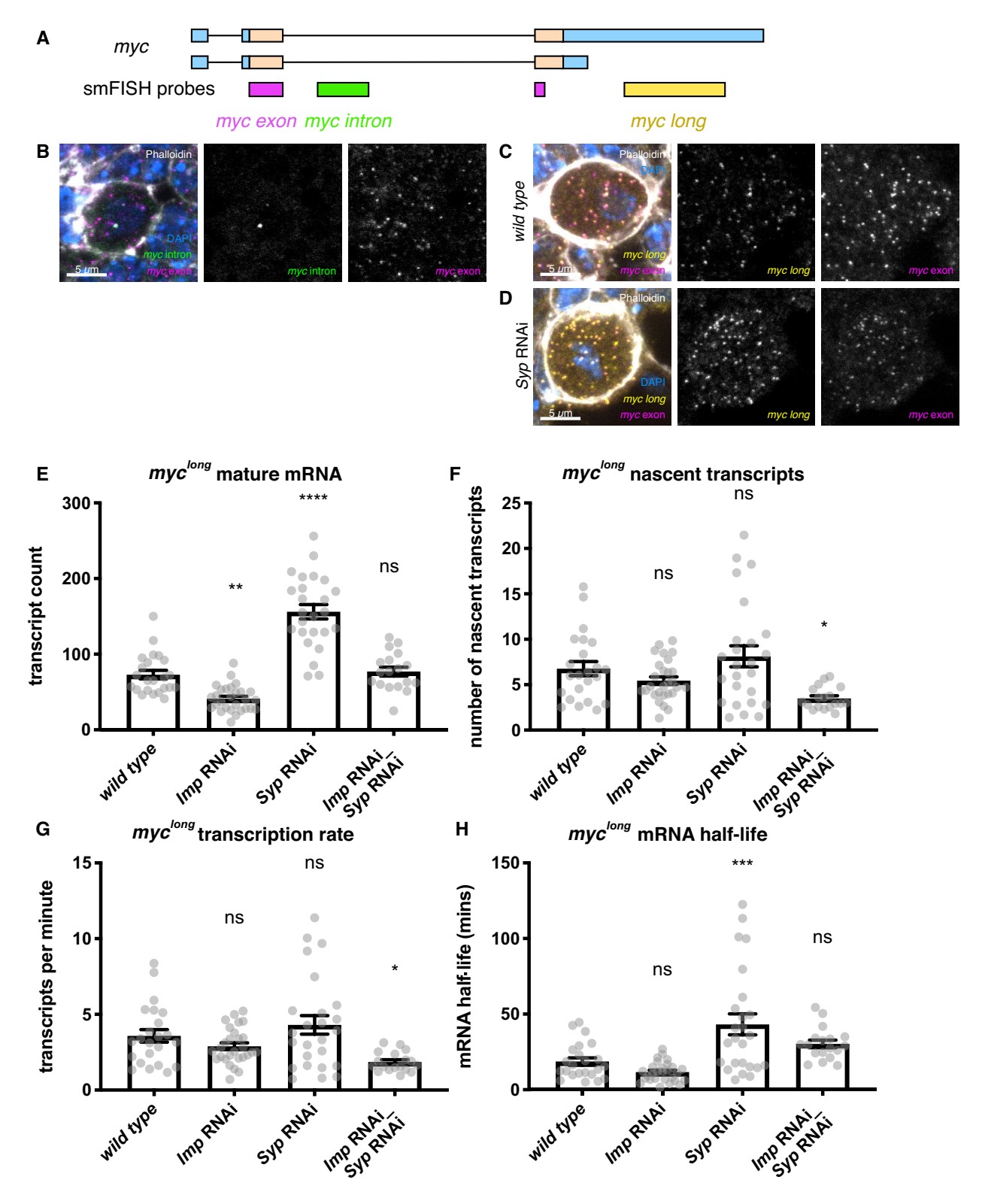

**Figure 4.** Imp stabilises *myc* mRNA. (**A**) We designed smFISH probes targeting the common exon (spanning the exon junction due to insufficiently long single exons), the intron, and the extended 3' UTR. (**B**) smFISH against the *myc* exon and the intron shows that *myc* is transcribed in type I NBs. (**C**) smFISH using probes against the common exon and the 3' UTR extension of *myc* shows that the long isoform of *myc* is expressed in the type I NBs. (**D**) *myc* transcript number is increased in the *Syp* knockdown. Z projection of 5 z planes. (**E**) The number of $myc^{long}$ transcripts was counted in individual

*Figure 4 continued on next page*

*Figure 4 continued*

NBs. The transcript number increased in the *Syp* RNAi but was unchanged in the double *Imp* and *Syp* RNAi. (**F**) The number of nascent transcripts was calculated using the integrated intensity from the transcription foci spot. The number of nascent transcripts was not significantly changed between genotypes. The counts of nascent and mature transcripts were then used to calculate *myc*^long^ half-life and transcription rate (**Bahar Halpern and Itzkovitz, 2016**). (**G**) The *myc*^long^ transcription rate is reduced in the *Imp Syp* double knockdown. (**H**) *myc*^long^ mRNA is stabilised in the *Syp* RNAi but the half-life is unchanged in the *Imp Syp* double knockdown. Significance calculated by ANOVA and Dunnett's multiple comparisons test, with comparison to *wild type*. ns = non significant, *p<0.05, **p<0.01, ***p<0.001, ****p<0.0001. error bars represent s.e.m. Each grey point represents one NB and for each genotype wL3 brains were analysed from three experimental replicates.

The online version of this article includes the following figure supplement(s) for figure 4:

**Figure supplement 1.** Workflow of transcript counting in NBs.

transcription rates (Materials and methods, [**Bahar Halpern and Itzkovitz, 2016**]). We found that the half-life of *myc*^long^ is not significantly changed in the *Imp* knockdown, but is significantly increased in the *Syp* knockdown, in which Imp is upregulated (*wild type* = 18.6 mins, *Syp* RNAi = 43.2 mins) (**Figure 4H**). This increase in *myc*^long^ mRNA half-life is suppressed in *Imp Syp* double knockdown NBs, in which there is no significant difference compared to *wild type*. It is not surprising that the *Imp* knockdown has no effect on *myc* mRNA half-life when compared to *wild type* NBs, because Imp levels are very low in *wild type* type I NBs at the wL3 stage. We find that Imp's main direct role is to promote *myc*^long^ mRNA stability and this results in upregulation of Myc protein, which promotes NB growth and division.

To characterise the regulation of Myc in other cells in the type I NB lineage, we used smFISH to observe *myc* transcription and cytoplasmic transcripts in the whole lineage (**Figure 4B,C**, **Figure 4— figure supplement 1**). We found that while *myc* is transcribed and transcripts are present in all cells in the lineage, Myc protein is limited to the NB only (**Figure 3A**), suggesting that *myc* transcripts are translationally repressed in the progeny GMCs and neurons. The repression of Myc protein expression in the progeny cells was unaffected by manipulation of Imp and Syp levels, driven by *insc-GAL4* (**Figure 3B**), suggesting that these two RBPs are not responsible for translational regulation of *myc*. While in the type II NB lineage, Brat is thought to translationally repress *myc* in progeny cells (**Betschinger et al., 2006**), it is not known to act in the type I lineage. We conclude that Myc is regulated in the NB lineages by mRNA stability through Imp and by translation, perhaps through a different RBP.

## High Imp stabilises *myc* mRNA in mushroom body NBs

The gradient of Imp level decline with developmental age is different between different NB types (**Liu et al., 2015**; **Syed et al., 2017**; **Yang et al., 2017a**). Therefore, we used smFISH to explore whether *myc* mRNA is also differentially stable in distinct NB types. Imp level declines more slowly in MB NBs compared to the rest of the type I NBs in the central brain and higher Imp expression remains in the MB NBs at wL3 (**Liu et al., 2015**; **Yang et al., 2017a**). In each NB, we used smFISH to measure *myc*^long^ transcription, *myc*^long^ mRNA half-life and *myc*^long^ transcript number as well as NB size and Imp protein level (**Figure 5A**). We identified MB NBs by their elevated Imp expression (**Figure 5A,B**). We found that MB NBs are 1.5-fold larger than type I NBs (**Figure 5C**). The *myc* mRNA half-life is 2.5-fold higher in the MB NBs (type I NBs = 18.79 mins, MB NBs, 51.34 mins) (**Figure 5D**, Materials and methods), while *myc* transcription rate is slightly reduced in the MB NBs compared to the type I NBs (**Figure 5E**). Plotting these variables together shows clear differences between the type I NBs and MB NBs. While type I NBs show low Imp, unstable *myc* mRNA and small NB size, the MB NBs have higher Imp, more stable *myc* mRNA and larger NB size (**Figure 5F**). These results support our earlier finding that higher Imp promotes *myc* mRNA stability and NB growth and indicates that Imp is a key regulator of differences between different classes of NBs.

We also measured Myc protein levels and NB division rates in MB NBs and type I NBs, although these could not be multiplexed into the same images as the smFISH measurements. We found that Myc protein level is 1.4-fold higher in MB NBs compared to type I NBs (**Figure 5G**). Finally, we measured NB division rate by incubation with EdU, which showed that MB NBs have a faster division

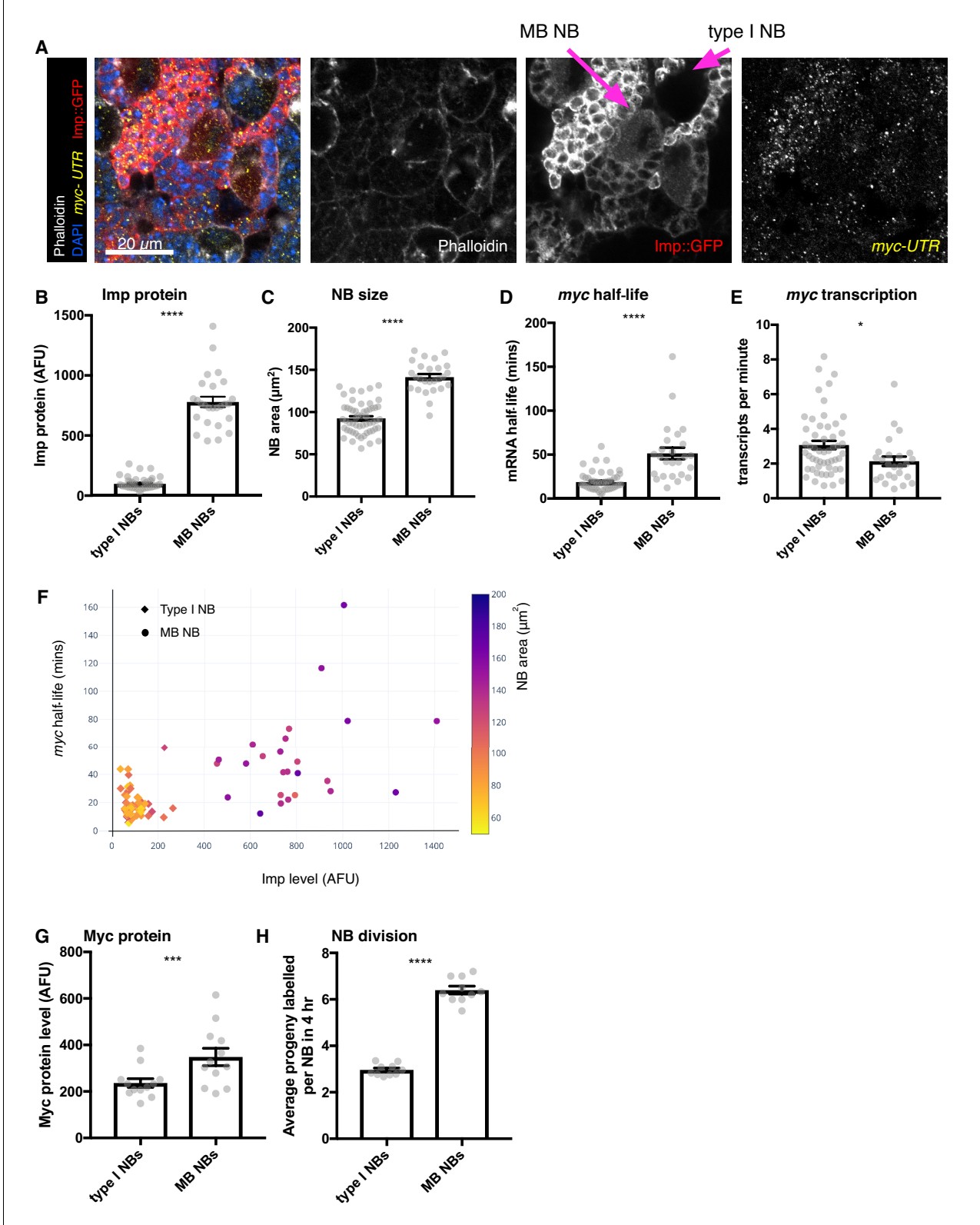

**Figure 5.** Higher Imp level in MB NBs leads to more stable *myc* mRNA. (**A**) wL3 brains expressing Imp::GFP and stained with *myc^long* smFISH probes and phalloidin were used to measure Imp level, NB size, *myc^long* transcription rate and half-life in individual NBs. MB NBs are identified by their higher Imp expression compared to type I NBs. (**B–D**) Each grey point represents one NB and for each NB type, brains were analysed from three experimental replicates. (**B**) MB NBs express higher Imp than type I NBs. The average intensity of cytoplasmic Imp signal is shown in arbitrary fluorescent units. (**C**)
*Figure 5 continued on next page*

*Figure 5 continued*

MB NBs are significantly larger than type I NBs, comparing NB area at the largest plane. (D) *myc* mRNA half-life is increased in MB NBs compared to type I NBs. (E) *myc* transcription rate is slightly lower in MB NBs than in type I NBs. (F) Plotting multiple measurements for each NB (Imp level against *myc* mRNA half-life, with NB size indicated by the colour scale) shows the differences between type I NBs (diamond point - low Imp, low *myc* mRNA stability, small) and MB NBs (circle point - high Imp, high *myc* mRNA stability, large). Imp level correlates with *myc* half-life. (G–H) Each grey point represents one brain and for each NB type, brains were analysed from three experimental replicates. (G) Myc protein is increased in MB NBs compared to type I NBs. (H) MB NBs produce more progeny in a four hour EdU incubation compared to type I NBs. Significance for each measurement was calculated using unpaired t-test, except for G) which uses a paired t-test. *p<0.01, ***p<0.001, ****p<0.0001.

rate than type I NBs (*Figure 5H*). Collectively, these results suggest that the higher level of Imp maintained into the late L3 stage in the MB NBs increases *myc* mRNA stability, causing increased Myc protein levels and increased NB growth and division relative to type I NBs at the same stage.

## Imp regulates *myc* mRNA stability throughout neuroblast development

Imp levels decline in NBs as larval development progresses (*Liu et al., 2015*) so we next asked what role Imp plays in *myc* regulation in earlier larval neurogenesis. We studied brains at 72 hr after larval hatching (ALH) when the Imp protein level in the NB is higher than at the later wL3 stage and there is substantial heterogeneity in Imp expression level between the individual NBs (*Figure 6A*). We first compared the average populations of 72 hr ALH NBs to wL3 NBs. Imp protein levels were measured from endogenous GFP-tagged Imp and found to be significantly increased in the 72 hr ALH NBs compared to wL3, as expected (*Figure 6B*). We then measured NB size and found that NBs are significantly larger at 72 hr ALH (*Figure 6C*). smFISH quantitation of $myc^{long}$ transcription and half-life at 72 hr ALH showed that $myc^{long}$ half-life is increased at 72 hr ALH (*Figure 6D*), but there was no significant difference in $myc^{long}$ transcription rate (*Figure 6E*). To validate the role of Imp in early larval neurogenesis, we measured NB size in Imp-depleted early NBs. NBs were much smaller in the *Imp* knockdown than in *Imp::GFP* (*wild type*) brains at 72 hr ALH (*Figure 6F*). This data supports the model that the decline in Imp levels during larval development reduces *myc* mRNA stability, restraining NB growth and division at the end of the larval stage.

Pooled averages hide the substantial variation in between individual NBs at 72 hr ALH so we asked whether the Imp level in each NB determines $myc^{long}$ half-life. We used a correlation matrix to examine the relationships between the variables measured in each individual NB at 72 hr ALH (*Figure 6G*, *Figure 6—figure supplement 1*) and found that Imp level correlates with $myc^{long}$ half-life (r = 0.344, p<0.01) in individual NBs. We also found a significant correlation between $myc^{long}$ transcript number and NB size (r = 0.281, p<0.05), which supports the hypothesis that Myc is a significant regulator of NB size at this stage. However, we found no significant correlation between Imp levels and $myc^{long}$ transcript numbers or NB size. The *myc* transcript number is controlled on multiple levels through both transcriptional and post-transcriptional mechanisms, and transcriptional activation of *myc* is a downstream consequence of many signalling pathways in the brain. Imp regulates *myc* mRNA stability to modify the final number of transcripts in each cell and as Imp levels decline through development *myc* mRNA stability also decreases. These results support the hypothesis that intrinsic Imp levels provide a mechanism to fine-tune the amount of Myc protein produced in each NB, allowing NB growth and division to be determined in each NB independently throughout its lifespan.

## Discussion

Each NSC produces a characteristic number of progeny to build a functional brain with the correct number of neurons of each type in each sub-region (*Yu et al., 2013*). However, how division rates are individually controlled through development is poorly understood. Here, we show that the temporally regulated RBPs Syp and Imp regulate NB division rate and size. Imp directly promotes NB growth and division through stabilising the mRNA of one of its key targets, *myc*, while Syp acts indirectly by negatively regulating Imp. By stabilising *myc* mRNA, Imp increases Myc protein expression

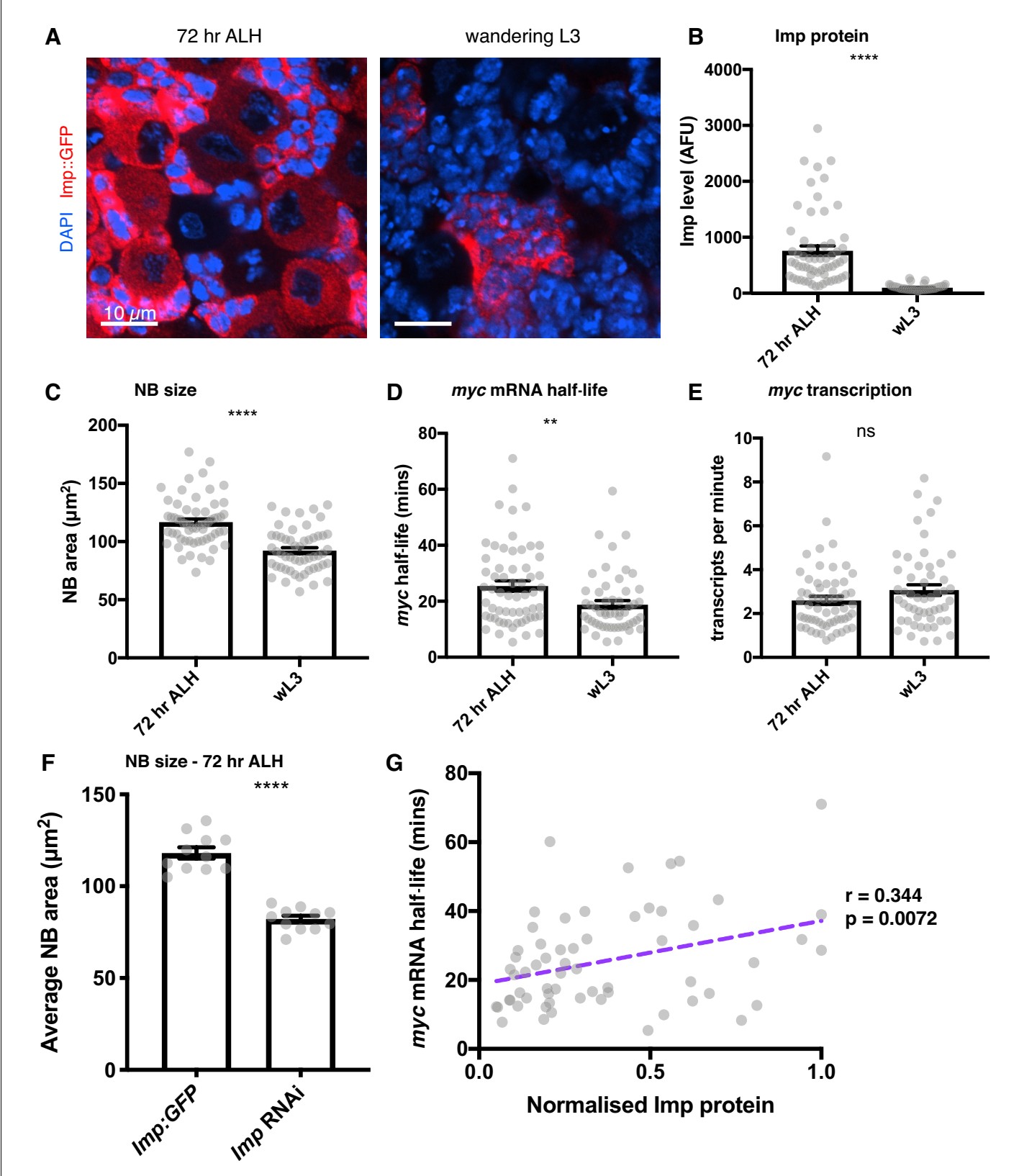

**Figure 6.** Imp stabilises *myc* mRNA throughout larval development. (**A**) Imp level (measured with endogenous Imp::GFP) is higher in NBs at 72 hr ALH compared to the wL3 stage, and is more variable between different type I NBs. Imp is very highly expressed in the progeny cells so the image is
*Figure 6 continued on next page*

*Figure 6 continued*

contrasted to show the Imp levels in the NBs. (B) Imp level quantitated in 72 hr ALH and wL3 type I NBs. (C) NBs are larger at 72 hr ALH compared to wL3. (D) *myc* mRNA half-life is longer in 72 hr ALH NBs compared to wL3. (E) The transcription rate of *myc* is not significantly different between 72 hr and wL3 NBs. Significance was calculated using unpaired t test. ns = not significant, **p<0.01, ****p<0.0001 F Measuring the size of type I NBs at 72 hr ALH shows *wild type* (*imp::GFP*) NBs are larger than *Imp* knockdown NBs. (G) In individual NBs at 72 hr ALH, increased Imp expression correlates with increased *myc* mRNA half-life. Imp level is normalised to the highest expressing NB from each imaging session. Each grey point represents one NB and for each stage, brains were analysed from three experimental replicates.

The online version of this article includes the following figure supplement(s) for figure 6:

**Figure supplement 1.** Imp regulates *myc* half-life in individual NBs at 72 hr ALH.

and drives NB growth and proliferation. Imp levels decline to low levels in type I NBs by the final wandering larval stage and we find that this results in low *myc* mRNA stability and low Myc protein levels. We show that Imp heterogeneity between NBs in earlier larval development (at 72 hr ALH), correlates with *myc* mRNA stability in individual NBs. Therefore, we suggest a model in which post-transcriptional regulation of *myc* mRNA stability by Imp provides a cell-intrinsic mechanism to fine-tune the growth and division rate of individual NBs, superimposed on the known extrinsic drivers of these processes (*Figure 7*).

## Post-transcriptional regulation of *myc* by Imp modulates NB growth and division

Myc is known to promote stem cell character and must be switched off in progeny cells to allow correct differentiation (*Betschinger et al., 2006*; *Gallant, 2013*). We found that Myc overexpression increases both type I NB size and division rate, which is a very interesting result since Myc is best known to drive cell growth through activation of ribosome biogenesis (*Grewal et al., 2005*). Myc also promotes a shortened G1 phase in the wing disc, but this does not increase division rate as the G2 phase is proportionately lengthened (*Johnston et al., 1999*). In the NB, the increased division rate we observe with Myc overexpression could be the result of a direct effect of Myc driving cell cycle progression, which would be mechanistically different from the cells of the wing disc. Alternatively, division rate may be increased indirectly as a result of the larger cell size. Further experiments will be required to uncover the precise mechanism of Myc action in the NB.

Our discovery of Imp-dependent modulation of Myc levels adds another dimension of regulation allowing cell-intrinsic modulation of NB growth and division tailored to individual NBs. It has been shown that Brat, an RBP, translationally represses Myc in type II NB progeny cells (intermediate neural progenitors) to prevent formation of ectopic NBs (*Bello et al., 2008*; *Betschinger et al., 2006*; *Boone and Doe, 2008*; *Bowman et al., 2008*). Together these findings emphasise the importance of the complex network of RBPs that play crucial post-transcriptional roles to control growth and division in individual NBs and their progeny in brain development.

Our work also suggests a new potential mechanism by which NB growth and division is restrained toward the end of the stem cell lifespan, in preparation for the terminal division in the pupa. The intrinsic regulation of *myc* mRNA stability by Imp could explain why NBs are insensitive to the general growth signalling pathways at their late stages (*Homem et al., 2014*). Homem et al., show that activation or inhibition of signalling through insulin-like peptides or their effector FOXO, has no effect on NB shrinkage or termination. Our results demonstrate that in the late larval NBs, there is insufficient Imp to stabilise *myc* mRNA, so that upregulation of *myc* transcription would still lead to low levels of Myc protein.

## Regulated Imp levels control *myc* mRNA stability in individual NBs and NB types

MB NBs are the longest lived NBs in the larval brain and their growth and division only finally slows at about 72 hr after pupal formation (*Siegrist et al., 2010*), 24 hr after the termination of the other type I NBs (*Yang et al., 2017a*). It was previously shown that NB decommissioning is initiated through a metabolic response to ecdysone signalling, via Mediator (*Homem et al., 2014*). Elevated Imp level inhibits Mediator in the MB NBs to extend their lifespan by preventing NB shrinkage (*Yang et al., 2017a*). However, *Yang et al. (2017a)*, found that inhibition of the Mediator complex

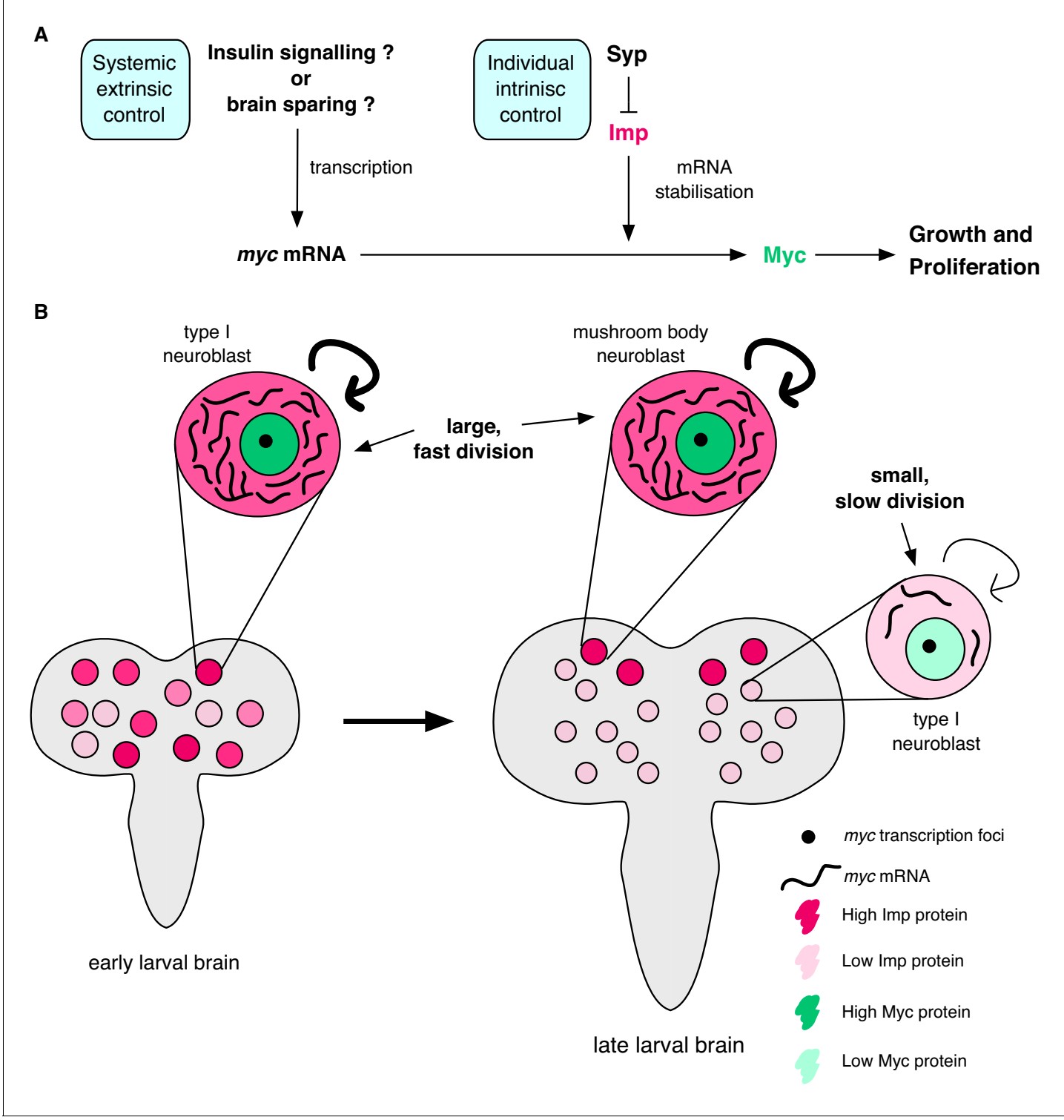

**Figure 7.** Imp stabilises *myc* mRNA to promote NB growth and division. (**A**) Myc drives growth and proliferation in NBs. We show that Myc level is regulated by intrinsic levels of Imp through increased *myc* mRNA half-life. Syp negatively regulates Imp to affect Myc levels indirectly. In our model, the post-transcriptional regulation of *myc* by Imp overlays potential extrinsic growth signals (labelled with a '?'), activating *myc* transcription. Multiple layers of regulation control growth and proliferation in each NB through development. (**B**) In early larval brains, Imp level is high, *myc* mRNA is relatively more stable and NBs are large. In individual NBs Imp level correlates with *myc* mRNA half-life. At the wandering larval stage Imp level is low in type I NBs, *myc* mRNA is unstable and NBs are small and divide slowly. This is in contrast to the MB NBs which maintain higher Imp levels, have more stable *myc* mRNA, and are larger and faster dividing.

only partially explained the lack of cell shrinkage in the long-lived MB NBs, suggesting that other targets of Imp also play a role in MB NBs. Imp stabilisation of *myc* mRNA might additionally promote NB growth to contribute to extending the MB NB proliferative lifespan. In contrast, Imp levels decline faster in the other type I NBs, which would restrain their growth and division in preparation for their earlier decommissioning.

We also examined the role of Imp earlier in larval development, at 72 hr ALH when Imp levels are higher and heterogeneous between individual NBs. Type I NBs at 72 hr ALH have higher *myc* mRNA stability and increased cell size compared to type I NBs at wL3. Our measurements of multiple variables in single cells allowed us to examine the function of Imp expression heterogeneity between individual NBs. We found that Imp levels correlate with *myc* mRNA stability in individual NBs at 72 hr ALH, providing a cell intrinsic mechanism to modulate NB growth and division. However, Imp levels do not correlate with NB size, unlike at the later wL3 stage. In the early larva, Imp and Myc levels are rapidly changing so a snapshot measurement of NB size may not be a suitable proxy for cell growth at each time point. Resolving this issue will require more sophisticated methods for long-term imaging of live whole brains that allow direct measurement of the growth and division rates of each NB at the same time as the Imp and Myc levels.

We have identified a mechanism of cell-intrinsic regulation of individual NB division and growth, which we suggest plays a key role in ensuring the correct number of progeny is produced in each lineage to build the correct sub-regions and circuits in the brain. This intrinsic regulatory mechanism must be integrated with extrinsic growth signals in the brain to determine the growth and division of each stem cell throughout development. Systemic insulin and ecdysone signalling are known to promote the timing of developmental switches in NBs, at the exit from quiescence after larval hatching and the decommissioning of the NB in the pupa. In the final stages of larval development, brain growth is also driven locally to protect it from nutrient restriction, in a process called brain sparing, by which Jelly-Belly expressed by the glial niche bypasses the insulin signalling pathway (*Cheng et al., 2011*). It is plausible that this local extrinsic regulation might also be specific to individual NBs, for example through controlled expression level of Jelly-Belly in each glial niche. Future experiments will determine the interplay between the intrinsic regulation of *myc* stability by Imp that we have shown here, and other extrinsic systemic and local regulators of NB growth and division.

## Declining Imp may restrain proliferation in diverse stem cell populations and systems

*c-myc,* the mammalian homologue of *Drosophila myc*, is best known for its role in cancers, and so its regulation has been studied extensively (reviewed in *Conacci-Sorrell et al., 2014*; *Farrell and Sears, 2014*). It is therefore interesting to consider to what extent the mechanism we have uncovered is conserved between *c-myc* and *Drosophila myc*. The mammalian homologue of Imp, IGF2BP1, binds to *c-myc* mRNA and regulates its stability. However, IGF2BP binds to *c-myc* mRNA in the coding sequence, whereas Imp binds to *myc* UTRs in *Drosophila*. IGF2BP1 is known to stabilise *c-myc* transcripts by blocking translation-coupled decay (*Bernstein et al., 1992*; *Doyle et al., 1998*; *Lemm and Ross, 2002*; *Weidensdorfer et al., 2009*), but in *Drosophila,* Imp's exact mechanism of stabilisation is not yet known. Nevertheless, the similarity of the two cases suggests that Imp regulation of *myc* stability might play a conserved role, coordinating stem cell growth and division with developmental progression.

The activity of stem cells in every context must be precisely restrained to prevent uncontrolled proliferation, and produce the correct numbers of each cell type to build the organ. We have discovered an important new regulatory mechanism, that Imp acts through *myc* mRNA stability to modulate cell growth and division appropriately in each stem cell and each stage of development. During development, lengthening of the G1 phase to extend the cell cycle length of NSCs is correlated with a switch from expansion to differentiation in the mouse ventricular zone (*Takahashi et al., 1995*). It has been proposed that Myc is a critical link between cell cycle length and pluripotency (*Singh and Dalton, 2009*). In parallel, Imp expression levels have been shown to occur in declining temporal gradients in diverse stem cells including the *Drosophila* testis (*Toledano et al., 2012*) and, in vertebrates, mouse foetal NSCs (*Nishino et al., 2013*). These diverse studies support our proposal of a new general principal that Imp temporal gradients limit stem cell proliferative potential towards the end of their developmental lifespan, by reducing *myc* mRNA stability and leading to low Myc protein

level. Future experiments in a wide range of other organs and systems will now be required to test our model, and to examine the extent of Imp expression heterogeneity in other stem cell systems.

# Materials and methods

## Key resources table

| Reagent type or resource | Designation | Source or reference | Identifiers | Additional information |
|---|---|---|---|---|
| Gene (*Drosophila melanogaster*) | Syncrip (Syp) | | FBgn0038826 | |
| Gene (*Drosophila melanogaster*) | IGF-II mRNA-binding protein (Imp) | | FBgn0285926 | |
| Gene (*Drosophila melanogaster*) | Myc | | FBgn0262656 | |
| Gene (*Drosophila melanogaster*) | Mnt | | FBgn0023215 | |
| Genetic reagent (*D. melanogaster*) | *wild type* OregonR | Bloomington | | |
| Genetic reagent (*D. melanogaster*) | *Syp* RNAi | VDRC | VDRC 33011 | *;P(GD9477)v33011* |
| Genetic reagent (*D. melanogaster*) | *Imp* RNAi line | Bloomington | BL 34977 | *y(1) sc[*] v(1); P{y[+t7.7] v[+t1.8]=TRiP. HMS01168}attP2* |
| Genetic reagent (*D. melanogaster*) | Imp OE line UAS-Imp-RM-FLAG | *Liu et al., 2015* | | |
| Genetic reagent (*D. melanogaster*) | Myc OE line | FLY-ORF collection | F001801 | *M{UAS-Myc.ORF.3xHA.GW}* |
| Genetic reagent (*D. melanogaster*) | *Myc* RNAi | Bloomington | BL 54154 | *y(1) v(1); P{y[+t7.7] v[+t1.8]=TRiP. HMC03189}attP40* |
| Genetic reagent (*D. melanogaster*) | Imp::GFP | *Toledano et al., 2012* | | *Imp[CB04573]* |
| Genetic reagent (*D. melanogaster*) | insc-GAL4 | *Betschinger et al., 2006* | | |
| Antibody | α-Syncrip (guinea pig, polyclonal) | *McDermott et al., 2014* | | 1:2000 WB,1:500 IF |
| Antibody | α-GFP (rat, monoclonal) | Chromotek | 3H9 RRID:AB_10773374 | 1:1000 WB |
| Antibody | α-αTubulin (mouse, monoclonal) | Sigma | | 1:500 WB |
| Antibody | α-Imp (rabbit, polyclonal) | Gift from P. M. Macdonald | | 1:600 IF |
| Antibody | α-Deadpan (rat, monoclonal) | abcam | 11D1BC7 RRID:AB_2687586 | 1:200 IF |
| Antibody | α-Myc (mouse, monoclonal) | Gift from R. N. Eisenman and DSHB | P4C4-B10 | 1:100 IF |
| Antibody | α-Mnt (mouse, monoclonal) | Gift from R. N. Eisenman | | 1:100 IF |
| Commercial assay, kit | GFP-trap agarose beads | Chromotek | gta-20 | |

*Continued on next page*

*Continued*

| Reagent type or resource | Designation | Source or reference | Identifiers | Additional information |
|---|---|---|---|---|
| Commercial assay, kit | Stellaris DNA probes | Stellaris | | |
| Commercial assay, kit | Phalloidin | Sigma | | |
| Commercial assay, kit | RNAspin Mini kit | GE Healthcare | | |
| Commercial assay, kit | NEBNext Poly(A) mRNA Magnetic Isolation Module | NEB | | |
| Commercial assay, kit | Ion Total RNA-Seq Kit v2 for Whole Transcriptome Libraries | Life Technologies | | |
| Commercial assay, kit | Agilent High Sensitivity DNA Kit | Agilent | | |
| Commercial assay, kit | Click-iT EdU Alexa Fluor 488/594 Imaging Kit | Invitrogen | | |
| Software, algorithm | GraphPad Prism version 7 | GraphPad Software | | |
| Software, algorithm | ImageJ version 2.0.0 | Fiji | | |
| Software, algorithm | FISHquant | *Mueller et al., 2013* | | |
| Software, algorithm | Transquant | *Bahar Halpern and Itzkovitz, 2016* | | |

## Experimental model and subject details

*Drosophila melanogaster* fly stocks were kept at 18°C, but transferred to 25°C for crosses and experimental use. OregonR was the *wild type* strain. Flies were raised on standard cornmeal-agar medium.

## Method details

### RNA extraction

Third instar larval brains were dissected in Schneider's insect medium and then flash frozen in liquid nitrogen. Brains were homogenised using a pestle in IP buffer (50 mM Tris-HCl pH 8.0, 150 mM NaCl, 0.5% NP-40, 10% glycerol, one mini tablet of Complete EDTA-free protease inhibitor and 2 µl RNAse inhibitor (RNAsin Plus RNase Inhibitor, Promega). RNA was extracted using the RNASpin Mini kit (GE Healthcare) according to manufacturer's instructions.

Reverse transcription and quantitative PCR cDNA was produced from extracted RNA using RevertAid Premium Reverse Transcriptase (Thermo Fisher Scientific) according to manufacturer's instructions with the addition of 1 µl RNAse inhibitor (RNAsin Plus RNase Inhibitor, Promega).

Real time quantitative PCR was performed using primers specific to a transcript of interest, and where possible spanning an exon junction. qPCR was performed using SYBR Green Master Mix with the CFX96 Touch Real-Time PCR Detection System (BioRad). Cycle threshold (C(T)) values were calculated from the BioRad CFX software using a second differential maximum method. Input samples were used for a dilution series and the percentage input of each gene was calculated in the IP samples as a measure of pulldown. For primer sequences see *Table 1*.

### RNA immunoprecipitation

Wandering larval brains were dissected and homogenised in IP buffer (see RNA extraction). Input samples were taken. Each experiment was done in triplicate. 200 *Imp::GFP* brains were used per IP for sequencing. The lysate was incubated with GFP-Trap agarose beads (Chromotek) at 4°C for two hours and the unbound supernatant was collected. Beads were washed in cold IP buffer for 4x quick washes. The bound material was eluted by incubation for 30 min at 65°C in Elution buffer (50 mM Tris HCl (pH 8), 10 mM EDTA, 1.3% SDS, protease inhibitor, RNase inhibitor). The elution step was

**Table 1.** qPCR primers.

| Gene | Forward | Reverse |
|------|---------|---------|
| rp49 | GCTAAGCTGTCGCACAAA | TCCGGTGGGCAGCATGTG |
| pros | TATGCACGACAAGCTGTCACC | CGACCACGAAGCGGAAATTC |
| chic | CTGCATGAAGACAACACAAGC | CAAGTTTCTCTACCACGGAAGC |
| syp | TATGTGCGAAATCTTACCCAGGA | CGTTCCACTTTTCCGTATTGCTC |
| myc | CGGCAGCGATAGCATAAAAT | ACCTCGTCGGTAAGACTGTGA |
| Eip93F | cgatgtgaagtccgtcagag | gatttccgggcatctagctt |
| mamo | ccatcagagcccataaggtg | caaaacggacgtccttcaat |

repeated and the supernatants were pooled. RNA was extracted for IP samples and inputs and used for RT-qPCR or sequencing libraries.

## Western blot

Proteins were separated by SDS-PAGE on a 4–12% Novex gradient gel then transferred to nitrocellulose membrane with the Trans-Blot Turbo Transfer System (BioRad). Membranes were blocked in 50% Odyssey Blocking Buffer in 0.3% PBST (1x PBS with 0.3% Tween) for 1 hr at RT. The membrane was incubated with primary antibody overnight at 4°C. After rinsing, the membrane was incubated with fluorescently labelled secondary antibodies for LICOR (1:2000) for 2 hr at RT. Membranes were washed in 0.3% PBST and imaged with the LI-COR Odyssey.

## polyA selection

For RNA sequencing, after RNA extraction mRNA was enriched through polyA selection with the NEBNext Poly(A) mRNA Magnetic Isolation Module (NEB) according to manufacturer's instructions. Briefly RNA sample was added to washed beads with Binding buffer. Samples were incubated at 65°C for 5 min and then cooled to 4°C for RNA binding. Beads were washed in Wash Buffer and RNA was eluted at 80°C for 2 min. Binding, washing and elution steps were repeated to improve purification with final elution in 17 µl of Tris Buffer.

## RNA sequencing

Three biological replicates (n = 3) were produced for each sample (whole transcriptome/input or immunoprecipitation). Poly(A) enriched RNA was then used for library production using the Ion Total RNA-Seq Kit v2 for Whole Transcriptome Libraries (Life Technologies). Libraries were produced according to the Ion Total RNA-Seq Kit v2 protocol. Following quality control steps, adaptors were hybridised to the RNA fragments and RT reaction was performed followed by cDNA amplification with Ion Xpress RNA Barcode primers. Prior to sequencing, quality of cDNA libraries were assessed using Agilent High Sensitivity DNA Kit with the Agilent 2100 Bioanalyser. Libraries were pooled to a total concentration of 100 pM, with three samples multiplexed per chip. Sequencing was performed on an in house Ion Proton Sequencer, using the Ion PI IC 200 Kit (Life Technologies). Ion PI chips were prepared following manufacturer's instructions and loaded using the Ion Chef System.

## Staining and imaging

### Antibody staining for immunofluorescence (IF) in larval brains

Larval brains were carefully dissected in Schneider's medium and collected into 0.2 ml PCR tubes. Samples were rinsed once with 0.3% PBSTX (0.3% Triton-X in 1x PBS) and then fixed in 4% paraformaldehyde (PFA) (4% PFA in 0.3% PSTX) for 25 min (for wL3) or 15 min (for 72 hr ALH) at room temperature (RT). Samples were rinsed briefly 3x in 0.3% PBSTX, and then washed 3 × 15 min in 0.3% PBSTX at RT. Blocking was for 1 hr at RT in Blocking Buffer (1% bovine serum albumin (BSA) in 0.3% PBSTX). Samples were incubated with primary antibody diluted in Blocking Buffer overnight at 4°C on a rocker (Note: we were unable to optimise Myc antibody staining in 72 hr ALH brains). Samples were rinsed and then washed 3 × 15 min in Blocking Buffer at RT. Alexa Fluor secondary antibody (Thermofisher) was added at 1:200 in Blocking Buffer and samples were incubated for 1 hr at RT in

the dark. Samples were rinsed briefly and then washed 3 × 15 min in 0.3% PBSTX at RT. For nuclear staining, DAPI (4',6-diamidino-2-phenylindole) was included at 1:500 in the second 15 min wash. Brains were mounted in VECTASHIELD anti-fade mounting medium (Vector Labs). Slides were either imaged immediately or stored at −20°C.

## Single molecule RNA fluorescent in situ hybridisation (smFISH) for larval brains

smFISH probes were designed using the Stellaris Probe Designer version 4.2. The sequences against which the probes were designed are shown in *Supplementary file 2*. Stellaris DNA probes were gently resuspended in 95 µl fresh TE buffer and 5 ul RNAse inhibitor (RNAsin Plus RNase Inhibitor, Promega), and frozen at −80°C in 10 µl aliquots. Dissected brains from male larvae were rinsed once with 0.3% PBSTX and then fixed in 4% PFA (in 0.3% PSTX) for 25 min (for wL3) or 15 min (for 72 hr ALH) at RT. Samples were rinsed briefly and then washed 3 × 15 min in 0.3% PBSTX at RT. Samples were washed for 5 min in Wash Buffer (10% deionised formamide (stored at −80°C) and 2x SSC in DEPC water) and then incubated with 250 nM Stellaris DNA probes in Hybridisation Buffer (10% deionised formamide, 2x SSC and 5% dextran sulphate in DEPC water) overnight at 37°C on a rocker. Samples were rinsed briefly 3x in Wash Buffer, and then washed 3 × 15 min in Wash Buffer at 37°C. For nuclear staining DAPI (4',6-diamidino-2-phenylindole) was included at 1:500 in the second wash. Brains were mounted in VECTASHIELD anti-fade mounting medium (Vector Labs). Slides were either imaged immediately or stored at −20°C.

### Additional stains

DAPI was used to stain nuclei, and was added at 1:500 in one of the final wash steps before mounting. Phalloidin was used to label F-actin and was added in one of the final wash steps and incubated for 1 hr at 37°C. Fluorescein 488 phalloidin was used at 5 µl per 100 µl, 647 Phalloidin was used at 2.5 µl per 100 µl.

### 5-ethynyl-2'deoxyuridine (EdU) labelling

Brains were dissected in Schneider's medium and then transferred to Brain Culture Medium (80% Schneider's medium, 20% fetal bovine serum (Gibco ThermoFisher), 0.1 mg/ml insulin (Sigma)) with 25 µM EdU for 4 hr. Brains were then washed with Schneider's medium and fixed for 25 min in 4% PFA in 0.3% PBSTX at RT. The samples were rinsed and then washed 3 × 15 min in 0.3% PBSTX at RT before blocking for 1 hr at RT in Blocking Buffer. Samples were incubated with anti-Dpn antibody in Blocking Buffer overnight at 4°C. The following day, samples were washed in Blocking Buffer and then incubated with Alexa Fluor secondary antibody (Thermofisher) at 1:200 in Blocking Buffer and samples were incubated for 1 hr at RT in the dark. Samples were washed 3 × 15 min in 0.3% PBSTX at RT and then fixed in 1% PFA in 0.3% PBSTX at RT for 15 min. Samples were washed and then incubated in Blocking Buffer for 1 hr. The Click-iT reaction was carried out with the Click-iT EdU Alexa Fluor 488 Imaging Kit (Invitrogen) following manufacturer's instructions for 30 min at RT. Samples were washed in 0.3% PBST with 5 mM EDTA, once including DAPI, and then mounted in VECTASHIELD anti-fade mounting medium (Vector Labs). Samples were imaged on the same day.

### Image acquisition

An inverted Olympus FV3000 Laser Scanning Microscope was used for fixed imaging of larval brains. Images were acquired using 60x/1.30 NA Si UApoN objective. For smFISH quantitation images, pixel size was 74 nm in x and y, and 200 nm in z.

## Quantification and statistical analysis

### Image analysis

#### Replicates

For all imaging experiments, staining and imaging was performed in three technical replicates (i.e. staining on three independent days). For each replicate the number of brains analysed ranged from 1 to 5 depending on availability of larvae. These are biological replicates. In *Figures 1*, *3* and *5G– H*, the individual replicates are shown on all plots as individual points. In *Figures 5B–D* and *6*, the individual NBs measured are shown as individual points on the plots.

## Measuring NB size

We measured all type I NBs in the central brain on the ventral side. We used phalloidin staining to mark the NB cell boundary and the area at the widest z plane was manually measured using ImageJ. NBs undergoing mitosis were excluded. They were identified using Dpn staining, which is weak throughout the cell when the nuclear envelope has broken down during mitosis. In *Figure 1* the average NB size per brain is plotted.

## Measuring proliferation rates

We measured all type I NBs in the central brain on the ventral side. Proliferation rate was measured with EdU labelling of progeny cells. The number of EdU +ve progeny per NB (labelled with Dpn) were counted manually. In *Figure 1* the average number of progeny per NB in each brain is plotted.

## NB segmentation

Using ImageJ, single NBs were cropped and substacks were made to span the depth of each NB. The phalloidin staining was used to create a mask with the FIJI plugin MorphoLibJ, using the morphological segmentation feature (*Legland et al., 2016*). NBs undergoing mitosis (condensed chromatin in the DAPI channel) were excluded.

## smFISH

After segmentation as above, transcripts outside the NB boundary were removed. FishQuant (*Mueller et al., 2013*) was used in batch mode to count spots and calculate nascent transcripts using the integrated intensity calculation. In brief, an outline was produced for each NB, identifying the transcription focus (note that as *myc* is on the X chromosome, only male larvae were dissected so there was one transcription focus per NB). Transcription foci were easily identified as the largest spot in the nucleus, with relatively more signal from the more 5' exon probe compared to the 3' UTR probe. A single NB was analysed to set up the detection settings which were then applied in the batch mode of all NBs from each technical replicate. The filters were modified manually to optimise transcript detection, and then an average transcript was calculated from the entire batch and used to calculate the nascent transcript number.

We applied the method established by *Bahar Halpern and Itzkovitz (2016)* to convert transcript counts to rates of transcription and mRNA decay. Simply, the nascent transcript number can be used to estimate the transcription rate in each cell, accounting for the position of the probe along the transcript, and an estimated rate of transcriptional elongation. The rate of elongation (v) was estimated at 1.5 kb per minute, based on a variety of methods in different *Drosophila* tissues, which gave measurements from 1.1 to 1.5 kb/min (*Ardehali and Lis, 2009*). A probe library weighting factor was calculated using the TransQuant software to account for the position of the probe set along the gene (*Bahar Halpern and Itzkovitz, 2016*). For *myc long* smFISH probes, this factor was 0.15264. Assuming a steady state, where transcription equals mRNA degradation, the estimated transcription rate can then be used to calculate an estimate of mRNA half-life in each cell.

Transcription and decay rates were calculated using the equations below. Decay rates were then converted to half-lives.

1. Transcription rate (mRNA/hr) = ((nascent transcript number/weighting factor) x elongation rate)/gene length
2. Decay rate (per hour) = (chromosome fraction x transcription rate x number of chromosome copies)/transcripts in the cell
3. Half-life (mins) = (ln2/decay rate) x 60

The calculation (*Bahar Halpern and Itzkovitz, 2016*) helps to unpick the differences in regulation of transcription or mRNA decay between different genotypes or cell types. However, the assumptions required for the method should be carefully considered in the interpretation of the results. The transcription rate calculation assumes a constant estimated transcription elongation rate without pauses or pulsing. The equations are based on a steady state but, while we excluded NBs undergoing mitosis, a dividing cell like the NB is unlikely to reach a true steady state.

## Statistical analysis

Statistical analysis was performed using Prism (GraphPad Software). For image analysis (smFISH and phenotypic analysis) involving three or more comparison groups (genotypes), one-way ANOVA was used to identify difference between the results of different phenotypes and the *wild type* value. Dunnett's multiple comparison test was then used to calculate significance values of each comparison. This applies to *Figures 1*, *3* and *4*.

For analysis involving only two comparison groups, unpaired t-tests was used (*Figures 5B–D,H* and *6*). For *Figure 5G,a* paired-t-test was used to compare the intensity of Myc protein directly between NB types in the same brains.

In *Figure 6—figure supplement 1* and *Figure 6G* a correlation matrix was produced, computing r for every pair of Y values with Pearson correlation coefficients.

The qPCR data (*Figure 2—figure supplement 1B*) was analysed with a comparison for each gene between the test and control pulldowns. The significance was calculated using t-tests with correction for multiple comparisons with the False Discovery Rate method, using an allowance of 5%.

## Bioinformatics methods

### Analysis of RNAseq and RIPseq

Reads from three Imp RIPseq libraries and three RNAseq libraries were mapped to the *D. melanogaster* genome (BDGP6.22.97) downloaded from ENSEMBL using the STAR aligner (2.5.3a) (*Dobin et al., 2013*). The aligned reads were then assigned to genes using htseq-count (0.11.2) (*Anders et al., 2015*). Imp RIPseq enrichment over baseline RNA expression (RNAseq) was calculated from gene counts after library size correction, and genes were ranked according to this ratio. We additionally used DESeq2 (1.24.0) (*Love et al., 2014*) to determine statistically significant difference between the RIPseq and RNAseq. Genes with very low abundance (those with total count of less than 10 across 3 RNAseq libraries) were ignored from ranking. Non-coding RNAs that overlap other genes were flagged up and not considered for *Figure 2*. This data is available in a tabular format in *Supplementary file 1*. To capture gene ontology (GO) terms linked to cell growth, neural development, and key regulatory processes, we extracted all GO terms using GO.db (3.8.2) (*Carlson, 2019*) and defined the following categories: cell growth (all GO terms that contain word 'cell growth'), cell size ('cell size'), cell division ('cell division'), cell cycle ('cell cycle'), neural development ('nervous system development', 'neurogenesis'), RNA binding ('RNA binding'), DNA binding ('DNA binding'). The GO terms falling under these categories are listed in *Supplementary file 1*. Gene-to-GO term mapping was extracted from Biomart using the R package biomaRt (2.40.4) (*Durinck et al., 2009*). The data was analysed in R with the help of the tidyverse suite of packages (1.2.1) (*Wickham, 2017*). R libraries rtracklayer (1.44.3) (*Lawrence et al., 2009*) and GenomicRanges (1.36.0) (*Lawrence et al., 2013*) were used to extract information from the annotation (.gtf) file and determine gene lengths and overlaps. The plots shown in *Figure 2* were made using ggplot2 (3.2.1) (*Wickham, 2016*). Further details of the analysis and code are available in *Source code 1*.

The *Hansen et al. (2015)* S2 *wild type* RNAseq (SRX751581, SRX751582) and Imp RIPseq (SRX751579, SRX751580) datasets were downloaded from the Short Read Archive (SRA) using SRA toolkit (2.9.3) (SRA Toolkit Development Team, http://ncbi.github.io/sra-tools/). The reads were mapped to *D. melanogaster* genome (BDGP6.22.97) using the STAR (2.5.3a). Read counts per gene were calculated using HTSeq-count (0.11.2). The *Hansen et al. (2015)* Imp iCLIP-seq (SRX751573, SRX751574) and PAR-iCLIP-seq (SRX751575, SRX751576) datasets were downloaded from SRA. Illumina sequencing adapters were trimmed off using cutadapt (1.10) (*Martin, 2011*) and the first five bases (corresponding to molecular barcodes) were removed from sequence and appended to read name. The reads were then mapped to the *D. melanogaster* genome (BDGP6.22.97) using STAR (2.5.3a). *xlsites* from the iCount pipeline (*Curk et al., 2019*) was used to determine the number of unique crosslinked sites (unique cDNA molecules) for any given position. iCount *peaks* was then used to call significant peaks and iCount *cluster* to cluster significant peaks. To make the gene track plots for myc (*Figure 2—figure supplement 1*), brain and S2 RNAseq were converted to strand-specific bedgraphs using bedtools (v2.28.0) (*Quinlan and Hall, 2010*). The visualisation was done with Bioconductor package Sushi (1.22.0) (*Phanstiel et al., 2014*). For the S2 iCLIP-seq, (confident) peaks and corresponding clusters are shown. Only one, representative replicate for each data type is shown.

## Data and code availability

The presented RNA sequencing data has been deposited with Gene Expression Omnibus (GEO), with accession number GSE140704. Further details of the analysis and code are available in *Source code 1*.

## Acknowledgements

We are grateful to the University of Oxford Micron imaging facility for help with advanced microscopy. Fly stocks and antibodies were kindly gifted by Tzumin Lee, Paul MacDonald and Robert Eisenman. We are grateful to Tzumin Lee and Francesca Robertson for their advice on experimental design and to Alfredo Castello, Jeffrey Lee, Mary Thompson and Dalia Gala for comments on the manuscript. TJS was funded by Wellcome Trust Four-Year PhD Studentship (105363/Z/14/Z) and Wellcome Investigator Award 209412/Z/17/Z. ID and AIJ were funded by Wellcome Trust Senior Research Fellowship 096144/Z/17/Z and Wellcome Investigator Award 209412/Z/17/Z. DIH was funded by University College London.

## Additional information

### Funding

| Funder | Grant reference number | Author |
| --- | --- | --- |
| Wellcome | 105363/Z/14/Z | Tamsin J Samuels |
| Wellcome | 096144/Z/17/Z | Aino I Järvelin<br>Ilan Davis |
| Wellcome | 209412/Z/17/Z | Tamsin J Samuels<br>Aino I Järvelin<br>Ilan Davis |
| University College London | | David Ish-Horowicz |

The funders had no role in study design, data collection and interpretation, or the decision to submit the work for publication.

### Author contributions

Tamsin J Samuels, Conceptualization, Data curation, Formal analysis, Validation, Investigation, Visualization, Methodology, Writing - original draft, Writing - review and editing; Aino I Järvelin, Data curation, Software, Formal analysis, Investigation, Visualization, Methodology; David Ish-Horowicz, Conceptualization, Writing - review and editing; Ilan Davis, Conceptualization, Supervision, Funding acquisition, Writing - review and editing

### Author ORCIDs

Tamsin J Samuels https://orcid.org/0000-0003-4670-1139
Aino I Järvelin https://orcid.org/0000-0002-1225-4396
David Ish-Horowicz https://orcid.org/0000-0001-5684-7129
Ilan Davis https://orcid.org/0000-0002-5385-3053

### Decision letter and Author response

Decision letter https://doi.org/10.7554/eLife.51529.sa1
Author response https://doi.org/10.7554/eLife.51529.sa2

## Additional files

### Supplementary files

• Source code 1. Analysis of Imp targets in the *D. melanogaster* larval brain. Details and code used for the bioinformatic analysis of the Imp RIPseq and RNAseq data presented in *Figure 2* and *Figure 2—figure supplement 1*.

- Supplementary file 1. Imp targets and GO terms used for categorisation. Table of Imp RIPseq targets: including read counts from three Imp RIPseq libraries and three RNAseq libraries, differential expression and GO analysis. GO terms falling under the following categories are listed: cell growth, cell size, cell division, cell cycle, neural development, RNA binding, DNA binding.
- Supplementary file 2. Stellaris Probes.
- Transparent reporting form

## Data availability

The presented RNA sequencing data has been deposited with Gene Expression Omnibus (GEO), with accession number GSE140704.

The following dataset was generated:

| Author(s) | Year | Dataset title | Dataset URL | Database and Identifier |
|---|---|---|---|---|
| Samuels TJ, Järvelin AI, Davis I | 2019 | Imp/IGF2BP levels modulate individual neural stem cell growth and division through myc mRNA stability | https://www.ncbi.nlm.nih.gov/geo/query/acc.cgi?acc=GSE140704 | NCBI Gene Expression Omnibus, GSE140704 |

The following previously published dataset was used:

| Author(s) | Year | Dataset title | Dataset URL | Database and Identifier |
|---|---|---|---|---|
| Hansen HT, Rasmussen SH, Adolph SA, Plass M, Krogh A, Sanford J, Nielsen FC, Christiansen J | 2015 | *Drosophila* Imp iCLIP identifies an RNA assemblage co-ordinating F-actin formation | https://www.ncbi.nlm.nih.gov/geo/query/acc.cgi?acc=GSE62997 | NCBI Gene Expression Omnibus, GSE62997 |

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
