## [Decision Letter]

**Acceptance summary:**

The Imp and Syp RNA binding proteins have been shown to play important roles in patterning the divisions of neuroblasts in *Drosophila* (and in vertebrates). This work demonstrates that one of the targets of Imp is the gene Myc, which thus explains why Imp needs to be down-regulated at the end of neuroblast lives in order to terminate them. Therefore, the temporal and spatial regulation of Myc controlled by Imp determines neuroblast growth and rate of division. The authors’ utilization of sophisticated mRNA detection allowed them to quantify mRNA in neuroblasts, and to understand how temporal patterning of neuroblasts regulates their proliferation. This will provide further support for the critical role of Imp (and Syp) in neuroblast division and the identification of further targets will likely also allow the authors to understand how these genes contribute to neural diversity.

**Decision letter after peer review:**

Thank you for submitting your article "Imp/IGF2BP levels modulate individual neural stem cell growth and division through *myc* mRNA stability" for consideration by *eLife*. Your article has been reviewed by Utpal Banerjee as the Senior Editor, by Claude Desplan as the Reviewing Editor, and three reviewers. The following individuals involved in review of your submission have agreed to reveal their identity: Chris Q Doe (Reviewer #1).

The reviewers have discussed the reviews with one another and the Reviewing Editor has drafted this decision to help you prepare a revised submission.

As you will see, the reviewers are positive about the paper but they together provided a long list of points that could make the paper even better. Below is the extensive list in the three reviews. However, and as it the policy of *eLife*, we would like you to be able to resubmit within a two month period and therefore we have listed below the experiments that we deem essential to improve the paper that could be made in the required time frame. After discussing with the other reviewers, reviewer #3 took the time to list all required experiments and this was approved by the other reviewers. As for the remaining comments, they only require text modifications/discussion.

Experiments:

- Stain for *myc (myc* Ab) in WT at 72hour and wandering larval brains

- UAS-Imp and UAS-Imp RNAi x type 1 Nb Gal4 at wandering stage: Stain for Myc and quantify Myc levels (Myc Ab)

- UAS-Imp RNAi x type 1 Nb Gal4 at 72hour stage: Stain for Myc

- Figure 4 – smFISH against Myc in UAS-Imp x nb Gal4 – This might take quite some time, but it would clarify the inconsistencies between the different genotypes. However, you might be able to argue your case if you feel that it is not essential

Data analysis:

- Correlation analysis of smFISH of type 1 NBs done separately from MB NBs. The same correlation analysis could be done for MBs.

- Figure 6F – show NB size in this graph to see whether there is correlation between *myc*/imp and cell size at the individual NB level. You might have the data already.

- Measure Myc levels and NB size from the *Imp* RNAi experiment:

UASImp RNAi x type 1 nbGal4 in 72hour ALH

- You should also remove the smFISH.

*Reviewer #1:*

This manuscript makes the following conclusions:

- Imp promotes NB size and rate of division

- Imp binds *myc* mRNA

- Imp binding stablizes *myc* mRNA

- Imp acts via *myc* to promote NB size and rate of division

- leading to a final model: Imp post-transcriptionally regulates *myc* mRNA stability to fine-tune individual NB size and division rate in their appropriate developmental context.

Overall the manuscript is clearly written with the data supporting the conclusions. The figures are also clear and illustrate the results well. There are no issues with statistics. This is an excellent manuscript.

*Reviewer #2:*

The *Drosophila* CNS is built upon generation of thousands of neurons and glia by a small population of asymmetrically dividing neural stem cells, called neuroblasts (NBs). The division properties of NBs have to be tightly spatio-temporally controlled to ensure that they each generate the correct repertoire and number of progeny. It is known that extrinsic and intrinsic cues are operating to control NB divisions, although the underlying mechanisms are not fully understood. Recent work has shown that the mRNA binding protein Imp is a key promoter of NB division during development, and that is needs to be silenced by temporal patterning during larval stages to ensure the timely termination of NB divisions during metamorphosis. However, Imp is known to regulate a large number of mRNAs but its mode of action in NBs is not clear. In this manuscript, Samuels and colleagues investigate in more details the role of Imp in NBs during larval stages and perform RIP-seq to identify mRNA targets. One of their top candidates is Myc. Using antibody and smFISH, they demonstrate that Imp promotes Myc expression via the stabilization of an mRNA isoform with a long 3'UTR. They show that NBs with high levels of Imp correlate with high levels of Myc, larger size and faster divisions. On the ground of other genetic experiments, they conclude that the temporal and spatial regulation of Myc via Imp determines NB growth and division rate during development.

The manuscript is convincing, with beautiful in situs allowing mRNA quantifications in NBs, and the conclusions are well supported by the experimental data. This study provides an additional layer of understanding of how NB intrinsic temporal patterning regulates NB proliferative properties during development. However, I am less convinced about the conceptual advances that it is bringing.

Indeed, in mammals, Myc is a well-known target of Imp/IGF2BP proteins (Noubissi et al., 2006; Noubissi et al., 2010; Doyle et al., 1998 etc) and the molecular mechanisms by which Imp/IGF2BP proteins promote Myc mRNA stabilization have been investigated in great details (Lemm and Ross, 2002; Weidensdorfer et al., 2009; Huang et al., 2018 etc).

Moreover, Imp/IGF2BP genes are known to be expressed in early cortical progenitors in mice where they promote their proliferation during embryonic development (Nishino et al., 2013; Yang et al., 2015), regulating temporal changes in stem cell properties, as it does in *Drosophila* NBs (Yang et al., 2017). And obviously Myc is famous for promoting cell growth and proliferation. Although the link between Imp/IGF2BPs and Myc had not been described in neural progenitors to my knowledge, this was kind of expected given the known regulatory interactions between these two genes in other tissues and cancers.

*Reviewer #3:*

In this manuscript the authors address by which molecular mechanism Imp and Syp regulate neuroblast growth and division restraining it or promoting it in different temporal windows. With this goal, the authors show that Imp/Syp levels are important regulators of neuroblast area and division rate. To understand how Imp regulates neuroblast size/division the authors perform RIPseq of Imp::GFP and find Myc RNA as a binding target of Imp in the brain. The authors also show that in the brain Imp binds myc longer isoform 3' UTR. The authors then want to analyze how Imp affects the stability of Myc mRNA. The authors show that at an earlier larval stage when Imp is more highly expressed, myc's half-life is increased, showing that during larval development there is a correlation between Imp levels and Myc half-life. Consistently they also show that in early larval stages/Imp-high neuroblasts are bigger than in wl3. Summing up the authors find evidence for a model where Imp binds and stabilizes Myc mRNA stabilizing it. Since Imp is temporally expressed, being high in earlier larval stages and low/absent in later larval stages, this could explain how this temporal gene regulates neuroblast size in different stages.

This work and final model are very interesting, however, there are several caveats in this work that need to be addressed. These caveats are mainly due to the inconsistencies of the genotypes used to increase or lower Imp levels. Although the authors have tools to directly increase Imp (UAS-Imp) or decrease Imp (Imp-RNAi) they mainly choose to analyse Syp-RNAi and *Imp* RNAi_*Syp* RNAi. Since, as the authors show and partially discuss, Syp seems to have additional functions other than repressing Imp, and therefore not all the conclusions drawn from these experiments can be taken. Bellow I point out figures/text sections where this occurs. Also, there are some characterizations missing regarding Imp and Myc levels in early vs. late larval stages. These are important and will greatly improve the work, since this will show that levels of Imp/Myc are indeed normally changing in different temporal windows and are therefore relevant in the wild-type developing brain.

Additionally, the authors make several claims, including in the Title of the manuscript that they found how growth and division of individual neuroblasts is regulated, however they in fact study how *imp/syp* regulate growth of type 1 neuroblasts as a group and make some comparisons to another group of neuroblasts, the mushroom body neuroblasts. The experiments showing Imp levels in individual neuroblasts, are well documented and very interesting, but they are not followed up, and in their current state they show no causality between individual levels of Imp and any phenotype. Therefore, all these "individual nbs" mechanistic claims should be removed, specially from the Title and Abstract.

Essential revisions:

Figure 1 and Figure 1—figure supplement 1

Figure 1—figure supplement 1:

Figure 1—figure supplement 1B – Upregulation of Imp in *Syp* RNAi not clear from image, also it is not quantified. From this image *Syp* RNAi does not seem to be upregulating Imp.

In WT – Imp staining pattern is at the NB membrane (not clear why it would be in glia – need to either show that these are from glia or tone it down) and in differentiated cells; in *Syp* RNAi – Imp levels seem to be overall decreased, but there seems to be more Imp at the NB cytoplasm; In Imp OE – which should be the same as *Syp* RNAi, Imp is again at the neuroblast membrane and in differentiated cells, but it does not show the same pattern as in *Syp* RNAi.

The authors say that *Syp* RNAi causes an increase in Imp cytoplasmic levels and imply that is the important pool of protein. However, although Imp OE does not cause an accumulation of Imp in the NB cytoplasm, ImpOE does cause a NB phenotype (Figure 1A'B'). So why the difference? The authors should use ImpOE directly and avoid unspecific effects caused by *Syp* RNAi as shown later on.

Figure 1—figure supplement 1A – Show levels of Imp with the antibody as in Figure 1—figure supplement 1A but for 72hour larvae. This is important for two reasons: (a) Imp is reported to be almost gone at 72hour larvae in type 1 neuroblasts and to be completely gone by W3IL (van den Ameele and Brand, 2019); (b) so that we can later assess if Myc levels at these two stages are indeed correlated with Imp levels. Without this comparison it is not possible to classify Imp and Myc levels as high or low.

Figure 3 and Figure 4

For the section where the authors explore the relationship of Imp/Myc, the authors do not always directly change Imp levels. In Figure 3 "Imp regulates Myc protein expression…" the authors never manipulate Imp directly (ImpOE or *Imp* RNAi), mainly showing how Myc is affected after changing the levels of Syp (*Syp* RNAi as a proxy for Imp overexpression). There are several issues with this approach, since, as discussed by the authors (Figure 4), *Syp* RNAi may cause other effects than on Imp. It would therefore be important to include in figures/results how directly changing Imp levels affects Myc.

Figure 3 – Show how Imp overexpression affects *myc* protein levels (e.g. as in graph in Figure 1A' where authors show that ImpOE causes an increase in nb area).

Figure 3 – Based on the results of Figure 4, where *Imp* RNAi_*Syp* RNAi does not cause a reduction in Myc Half-life and mature mRNA (i.e. *myc* long isoform stability) as is the case for *Imp* RNAi, but instead possibly causes a decrease in transcription, it would be useful to see *Imp* RNAi and ImpOE effect on *myc* protein levels.

Figure 4 – The effect of *Imp* RNAi_*Syp* RNAi on *myc*/transcription is not consistent with this double RNAi being the same as knocking down Imp. It suggests that *Syp* RNAi has an effect independent of Imp. The authors conclusion that the result of this double KD shows "that Imp rather than Syp is the primary regulator of the number of *myc^long^*..." is therefore not supported by the data and needs to be toned down. For this reason, it would also be useful to have ImpOE. Add which developmental stage these analyses were done to the legend.

Figure 4H – "we find that Imp's main direct role is to promote *myc^long^* mRNA stability and this results in upregulation of Myc protein, which promotes NB growth and division" however myc half-life is not reduced in *Imp* RNAi, nor in *Imp* RNAi_*Syp* RNAi, but NB size and division rate is reduced in both genotypes (Figure 1A'B'). The authors need to reformulate their conclusion and discuss these inconsistencies.

Figure 5

In Figure 5F authors show Imp levels and *myc* half-life. The authors claim in several sections, including in Title and Abstract, that "Imp/IGF2BP levels modulate individual neural stem cell growth…" however the authors do not show this in individual type 1 nbs or show any causality or even correlation. Just by looking at 5F there seems to be no correlation between Imp levels and nb area. Comparison between type 1 central brain and MB is interesting, but these are different cell types, and as shown previously, regulated differently. Therefore these 2 cell types combined cannot be used to conclude what the authors claim. The authors need to: (a) remove individual cell claims, (b) perform a correlation analysis of myc half-life/IMP levels/nb size in only type 1 nbs, the focus of this work.

Figure 5H -Even when type 1 NBs are overexpressing Myc (Figure 3E) they do not reach the average number of progeny labeled as for MB (mycOE type1~4 progeny cells labeled in 4 hours; MB NBs ~6 progeny cells labeled in 4 hours). Therefore, it does not seem that Myc levels are the main cause of this difference. The authors need to rephrase their conclusions when comparing type1 and MB neuroblasts, since these two nb types seem to have many other differences.

Include in figure legend (check all legends) what is the exact stage analysed.

Figure 6

In Figure 6 the authors try to address how the differential expression of Imp/Syp throughout time affect myc levels and neuroblasts. They find that at 72hour ALH, Imp::GFP levels are high in nbs and in differentiated cells, but at wandering L3 Imp::GFP levels are much reduced, and that this reduction correlates with a decrease in neuroblast area and myc half-life. However, the authors do not show protein levels of Myc (Myc staining) in these 2 stages, this should be included.

The authors should also knock down Imp in 72hour ALH, the stage where Imp is high, and therefore normally playing a role, and see how this affects Myc protein levels.

Figure 6F – show NB size too in this graph – to see if indeed there is correlation between myc/imp and cell size at the individual nb level.

Figure 7 – The insulin signaling or brain sparing is not connected with the process that the authors are studying, i.e. normal temporal regulation of neuroblasts. Remove from this figure as it is confusing. The "individual intrinsic control" is also not clear or supported by the data, remove.

Title

The authors are interested in understanding how Imp levels modulates individual neural stem cell growth, however the authors study in general type 1 neuroblasts and do not perform experiments to study how the growth of individual neuroblasts is regulated by Imp. There is one experiment (5F) that characterizes myc half-life, Imp levels and NB area in single neuroblasts, but this experiment does not show if there is correlation among type 1 nbs nor does it show if a difference in levels among type 1 nbs is causal of differences in neuroblast size. For these reasons the Title does not fit with what is really shown in the manuscript, which is how mechanistically Imp levels modulate neuroblast (type 1) growth and division through myc mRNA stability. This needs to be changed.

Abstract

The Abstract should also be adjusted to not highlight differences between individual neuroblasts. E.g. "mechanisms that control the characteristic proliferation rates of individual neural stem cells are unknown" – although correct, this is not what the authors address in their manuscript.

The authors say that "the division of neural stem cells, regulated systemically by known extrinsic signals." – "Some", or "Important" extrinsic signals are known.

Graphical abstract:

The graphical abstract should be adjusted according to comments above.

---

## [Author Response]

As you will see, the reviewers are positive about the paper but they together provided a long list of points that could make the paper even better. Below is the extensive list in the three reviews. However, and as it the policy of eLife, we would like you to be able to resubmit within a two month period and therefore we have listed below the experiments that we deem essential to improve the paper that could be made in the required time frame. After discussing with the other reviewers, reviewer #3 took the time to list all required experiments and this was approved by the other reviewers. As for the remaining comments, they only require text modifications/discussion.

We thank the reviewers and editors for distilling the essential points we need to address and agree that these changes have improved the paper considerably. In our revision we have focused on addressing the distilled points, but we have also attempted to address all the other comments, as in our view they also improve the manuscript.

Experiments:- Stain for myc (myc Ab) in WT at 72hour and wandering larval brains.

While we show Myc antibody staining in wandering larval brains in Figure 3A, we have been unable to reproducibly stain 72 hour larval brains with the Myc antibody. We have tried hard to carry out this experiment, both before and after receiving these review comments, by testing different fixing protocols (including methanol fixing and 1% or 4% formaldehyde) as well as different Myc antibody concentrations and pre-clearing the antibody. While some of our stainings indicate that Myc protein is quite highly expressed in 72 hour type I NBs, we are not comfortable to publish these data because the stainings are not reproducible and so can not be validly quantitated. Nevertheless, we do show in Figure 6 that NBs are larger at 72 hour compared to wL3 and that *myc* mRNA half-life is increased. Although we agree that the Myc staining would have been a nice addition, we do not think that it is essential to our argument that higher Imp at 72 hour results in increased *myc* mRNA stability, which we measure directly. We have added a short statement to the Materials and methods section saying that the Myc antibody does not work in our hands in 72 hour larvae.

- UAS-Imp and UAS-Imp RNAi x type 1 Nb Gal4 at wandering stage: Stain for Myc and quantify Myc levels (Myc Ab).

This was a very useful suggestion, thank you. The results have also helped us to address some of the additional specific comments raised by the reviewers. We have added measurements of Myc protein levels in the UAS-Imp and *Imp* RNAi lines to Figure 3C. The results show a small increase in Myc protein when Imp is overexpressed (OE). This effect is expected, as only a small Imp OE can be achieved with this construct (Figure 1 – —figure supplement 1D), as previously observed by Tzumin Lee’s lab (Liu et al., 2015; Yang et al., 2017). In the *Imp* RNAi, we observe only a small, reduction in Myc protein. This result is also expected because Imp levels are already very low in the *wild type* at wL3. These findings support our interpretation that Imp stabilises *myc* mRNA to determine Myc protein levels in the type I NBs, which in turn determine their rate of division and growth.

- UAS-Imp RNAi x type 1 Nb Gal4 at 72hour stage: Stain for Myc.

As addressed above, we have been unable to get reproducible specific Myc antibody staining at 72 hour ALH. However, we have knocked down Imp in type I NBs at 72 hour and instead measured NB size, showing that NBs are much smaller in the *Imp* RNAi, compared to the *wild type* at this stage (Figure 6F). Note that at the 72 hour Imp levels are high (compared with the very low third instar Imp levels described above, Figure 6A), explaining why the *Imp* RNA has large effect at the earlier stage. Again, these findings agree with our interpretation, namely that Imp promotes increased NB growth through stabilising *myc* mRNA.

- Figure 4 – smFISH against Myc in UAS-Imp x nb Gal4 – This might take quite some time, but it would clarify the inconsistencies between the different genotypes. However, you might be able to argue your case if you feel that it is not essential.

As described above, overexpression of Imp from a UAS construct results in minimal upregulation of Imp protein in the NB, and we now show that using the Imp OE construct we only see a small increase in Myc protein (Figure 3C). In response to the above comment, we have stained UAS-Imp brains with smFISH against *myc* mRNA (Figure 4—figure supplement 1B), but we do not observe a substantial upregulation of *myc* mRNA, especially compared to the increase seen with *Syp* RNAi. We think that quantitations of *myc* half-life and transcription rate in the Imp OE genotype will be much less useful than our studies using the *Syp* knockdown, which produces a very large upregulation of Imp protein in the NB and a corresponding large upregulation of Myc protein.

Data analysis:- Correlation analysis of smFISH of type 1 NBs done separately from MB NBs. The same correlation analysis could be done for MBs.

Based on reviewer 3’s comment below, we think this comment is referring to Figure 5F in which we plot *myc* mRNA half-life verses Imp levels and NB size for all the NBs. These data are also plotted in bulk in Figure 5B-E, but we show the individual NBs in a single graph in 5F to illustrate more clearly the differences between NB types. Identifying the different cell types to clearly separate the type I NBs and MB NBs is a very good suggestion. We have addressed this by adjusting the graph in Figure 5F to mark the type I NBs with a diamond and MB NBs with a circle, in order to clearly emphasise the differences between the NB types.

The reviewer also asks us to show do separate correlation analysis in each of the two different classes of NBs. However, we think that doing so would not be very informative for type I NBs because the at wL3, Imp levels in *wild type* type I NBs are very low (at the detection threshold). Therefore, we don’t think it is informative to look for correlations between Imp level (always very low) and *myc* mRNA stability in type I NBs at the wL3 stage. However, we did correlation analysis for MB NBs at wL3 using the data shown in Figure 5F, which is potentially useful as the MB NBs express higher Imp levels and there is variability between individual MB NBs. Although we have only very limited cell numbers for this type of analysis, we find that Imp level significantly correlates with *myc* transcript number (r = 0.48, p = 0.015). There is also a positive correlation between Imp level and *myc* mRNA half-life but this is not statistically significant (r = 0.29, p = 0.18). We have not gathered additional data for more individual MB NBs in this experiment in order to add it to the paper, as it would be very time consuming and we don’t think it is essential for the conclusions of the paper, which primarily focuses on type I NBs. In any case, we do already show in the paper the correlation analysis for type I NBs at stage 72 hour (Figure 6G, Figure 6—figure supplement 1), when there is much larger variation in Imp levels between individual type I NBs. We feel the data showing correlations in Figure 6 is strong and is sufficient for us to make our conclusions on individual NBs.

- Figure 6F – show NB size in this graph to see whether there is correlation between myc/imp and cell size at the individual NB level. You might have the data already.

The reviewer is correct: we do have all the data they are asking for, which is displayed in Figure 6—figure supplement 1 including all the correlation values and significance. We have measured 5 distinct variables, so we feel that it is better to represent them in a table showing a correlation matrix than to show plots of all the possible combinations. The reason we chose to show the one graph we did is that the correlation between Imp protein levels and *myc* mRNA stability in individual NBs is the key subject of this paper. Regarding the requested specific correlations between NB size and Imp/*myc*, we show in Figure 6—figure supplement 1, that *myc* transcript number does correlate with NB size, but Imp level does not correlate with NB size. We highlight this result in the corresponding Results section, and also discuss the reasons in the Discussion section. Cell size may be a poor proxy for cell growth at the early larval stage of 72hour ALH, when NB sizes are rapidly changing and we suggest in the Discussion section that long-term live imaging with Imp measurement could provide a better quantitation of cell growth.

- measure Myc levels and NB size from the Imp RNAi experiment:UASImp RNAi x type 1 nbGal4 in 72hour ALH.

In response to the above comment, we have measured NB size in the *Imp* RNAi at 72 hour ALH, and found that NBs are much smaller in the *Imp* RNAi, compared to the *wild type* at this stage (Figure 6F). This is a useful addition to the paper as it confirms that Imp drives NB growth at the earlier 72 hour stage, as well as at wL3 (Figure 1A, A’). As explained above, we were unable to produce Myc antibody staining at 72 hour ALH.

Reviewer #2:[…] The manuscript is convincing, with beautiful in situs allowing mRNA quantifications in NBs, and the conclusions are well supported by the experimental data. This study provides an additional layer of understanding of how NB intrinsic temporal patterning regulates NB proliferative properties during development. However, I am less convinced about the conceptual advances that it is bringing.Indeed, in mammals, Myc is a well-known target of Imp/IGF2BP proteins (Noubissi et al., 2006; Noubissi et al., 2010; Doyle et al., 1998 etc) and the molecular mechanisms by which Imp/IGF2BP proteins promote Myc mRNA stabilization have been investigated in great details (Lemm and Ross, 2002; Weidensdorfer et al., 2009; Huang et al., 2018 etc).Moreover, Imp/IGF2BP genes are known to be expressed in early cortical progenitors in mice where they promote their proliferation during embryonic development (Nishino et al., 2013; Yang et al., 2015), regulating temporal changes in stem cell properties, as it does in *Drosophila* NBs (Yang et al., 2017). And obviously Myc is famous for promoting cell growth and proliferation. Although the link between Imp/IGF2BPs and Myc had not been described in neural progenitors to my knowledge, this was kind of expected given the known regulatory interactions between these two genes in other tissues and cancers.

While it is true that in mammals the relationship between Myc and Imp has been previously studied, our imaging approach provides a major conceptual advance. We measure *myc* regulation in single cells within the developing brain to uncover how Imp regulates Myc to regulate individual neural stem cell size and division rate. Finally, in the Discussion section of the manuscript we address this point directly and reference much of the above mammalian literature.

Reviewer #3:[…] This work and final model are very interesting, however, there are several caveats in this work that need to be addressed. These caveats are mainly due to the inconsistencies of the genotypes used to increase or lower Imp levels. Although the authors have tools to directly increase Imp (UAS-Imp) or decrease Imp (Imp-RNAi) they mainly choose to analyse Syp-RNAi and Imp RNAi_Syp RNAi. Since, as the authors show and partially discuss, Syp seems to have additional functions other than repressing Imp, and therefore not all the conclusions drawn from these experiments can be taken. Bellow I point out figures/text sections where this occurs.

(Many of these points have been addressed in more depth above in our responses to the distilled list from the reviewers)

We have expanded on these points below, where they are individually raised, but briefly: we primarily use the *Syp* RNAi to upregulate Imp because the effect is much greater than using the UAS construct to directly overexpress Imp. We have added the experiments suggested by reviewer 3, measuring Myc protein level in the UAS-Imp and *Imp* RNAi brains, and find small changes in Myc level, as we would predict. This result is a useful addition to the paper, supporting our model. However, it remains the case, that the larger Imp upregulation in the *Syp* RNAi brains is a better system for us to detect changes in *myc* mRNA stability using smFISH.

Also, there are some characterizations missing regarding Imp and Myc levels in early vs. late larval stages. These are important and will greatly improve the work, since this will show that levels of Imp/Myc are indeed normally changing in different temporal windows and are therefore relevant in the wild-type developing brain.

We have measured Imp level in 72 hour and wL3 NBs and show that Imp level decreases in the NB over this developmental time period. However, despite trying hard, we have not been able to optimise the Myc antibody for staining 72 hour ALH brains and so unfortunately, we have not been able to compare Myc level at 72 hour and wL3. Although this was a relevant and useful suggestion, we do not think this experiment is essential for our conclusions that Imp promotes *myc* mRNA stability.

Additionally, the authors make several claims, including in the Title of the manuscript that they found how growth and division of individual neuroblasts is regulated, however they in fact study how imp/syp regulate growth of type 1 neuroblasts as a group and make some comparisons to another group of neuroblasts, the mushroom body neuroblasts. The experiments showing Imp levels in individual neuroblasts, are well documented and very interesting, but they are not followed up, and in their current state they show no causality between individual levels of Imp and any phenotype. Therefore, all these "individual nbs" mechanistic claims should be removed, specially from the Title and Abstract.

The reviewer is correct that for the first part of the paper we study type I NBs at wL3 as a group, as this is a good system to measure the effects of experimental manipulations of Imp level. We then compare type I NBs to mushroom body NBs, which have a higher Imp level at the same wL3 stage. However, at the end of the paper we measure Imp level, *myc* mRNA stability and NB size in individual type I NBs at 72 hour ALH (Figure 6G and Figure 6—figure supplement 1). At this stage there is heterogeneous Imp expression between individual type I NBs and so this is the ideal system to test the effect of different Imp levels in individual NBs. We find a significant positive correlation between Imp levels and *myc* mRNA stability in individual type I NBs at 72 hour ALH. Collectively, we feel that we have provided good evidence to support our model that Imp stabilises *myc* mRNA in individual NBs.

Essential revisions:Figure 1 and Figure 1—figure supplement 1Figure 1—figure supplement 1:Figure 1—figure supplement 1B – Upregulation of Imp in Syp RNAi not clear from image, also it is not quantified. From this image Syp RNAi does not seem to be upregulating Imp.

We have updated Figure 1—figure supplement 1 by zooming closer to the NBs to show the upregulation of Imp in the *Syp* RNAi. We also indicated a single type I NB in each image with an arrow. We feel it should be much easier for the reader to now see the upregulation of Imp.

In WT – Imp staining pattern is at the NB membrane (not clear why it would be in glia – need to either show that these are from glia or tone it down) and in differentiated cells; in Syp RNAi – Imp levels seem to be overall decreased, but there seems to be more Imp at the NB cytoplasm; In Imp OE – which should be the same as Syp RNAi, Imp is again at the neuroblast membrane and in differentiated cells, but it does not show the same pattern as in Syp RNAi.

To address this comment, we have stained a glial GAL4 (nrv2-GAL4) driving mCD9::GFP with Imp antibody and showed that Imp is expressed in glial cells. However, it is not directly relevant to the paper, so we have not included these data and instead altered the wording in the legend of Figure 1—figure supplement 1 to tone down the suggestion that Imp is expressed in glia. We agree with the reviewer’s observation that there is a reduction in Imp levels in many neurons in the *Syp* RNAi. However, we are measuring changes in only the NB where, as the reviewer observes, Imp is upregulated in the cytoplasm. The Imp OE is not very effective at upregulating Imp in the NB (described extensively above) and we agree that the greatest effect is in the neurons. For this reason, we primarily use the *Syp* RNAi to ‘upregulate’ Imp in the NB.

The authors say that Syp RNAi causes an increase in Imp cytoplasmic levels and imply that is the important pool of protein. However, although Imp OE does not cause an accumulation of Imp in the NB cytoplasm, ImpOE does cause a NB phenotype (Figure 1A'B'). So why the difference? The authors should use ImpOE directly and avoid unspecific effects caused by Syp RNAi as shown later on.

We agree that it would be cleaner to use the Imp OE directly. However, in agreement with previously published work, we find expressing Imp from an overexpression construct in NBs has minimal effect on Imp expression at wL3 (Liu et al., 2015; Yang et al., 2017) (Figure 1—figure supplement 1D). The Imp upregulation is much more dramatic in the *Syp* RNAi line and therefore we use the *Syp* RNAi line to upregulate Imp for the purposes of the smFISH quantitation (explanation added to subsection “Myc expression is regulated by Imp levels”). Nevertheless, we measured Myc protein expression in the Imp overexpression construct as suggested by the reviewer below, and we found a small increase in Myc protein levels in the NB. This small increase in Myc protein may be sufficient to explain the phenotypes that we see in the Imp OE in Figure 1.

Figure 1—figure supplement 1A – Show levels of Imp with the antibody as in Figure 1—figure supplement 1A but for 72hour larvae. This is important for two reasons: (a) Imp is reported to be almost gone at 72hour larvae in type 1 neuroblasts and to be completely gone by W3IL (van den Ameele and Brand, 2019); (b) so that we can later assess if Myc levels at these two stages are indeed correlated with Imp levels. Without this comparison it is not possible to classify Imp and Myc levels as high or low.

We show and measure the levels of Imp in 72 hour larvae and wL3 larvae in Figure 6A and 6B with the *Imp::GFP* line. In our hands in the *Imp::GFP* genotype, there is reproducibly Imp protein remaining at 72 hour ALH, and this is lost (below detection threshold) by wL3.

Figure 3 and Figure 4For the section where the authors explore the relationship of Imp/Myc, the authors do not always directly change Imp levels. In Figure 3 "Imp regulates Myc protein expression…" the authors never manipulate Imp directly (ImpOE or Imp RNAi), mainly showing how Myc is affected after changing the levels of Syp (Syp RNAi as a proxy for Imp overexpression). There are several issues with this approach, since, as discussed by the authors (Figure 4), Syp RNAi may cause other effects than on Imp. It would therefore be important to include in figures/results how directly changing Imp levels affects Myc.Figure 3 – Show how Imp overexpression affects myc protein levels (e.g. as in graph in Figure 1A' where authors show that ImpOE causes an increase in nb area).Figure 3 – Based on the results of Figure 4, where Imp RNAi_Syp RNAi does not cause a reduction in Myc Half-life and mature mRNA (i.e. myc long isoform stability) as is the case for Imp RNAi, but instead possibly causes a decrease in transcription, it would be useful to see Imp RNAi and ImpOE effect on myc protein levels.

Collectively, these comments ask for quantitations of Myc protein in the *Imp* RNAi and *Imp* OE type I NBs at wL3. This was a very useful suggestion, discussed fully above in the main response. Briefly, we have added the requested quantitations of Myc protein levels in the UAS-Imp and UAS-*Imp RNAi* lines at the wandering stage in Figure 3C, referred to in subsection “Imp binds hundreds of mRNA targets in the brain, including *myc*”. We find a small increase in Myc protein in the UAS-Imp NBs, which corresponds to the minimal upregulation of Imp in the Imp overexpression constructs (Figure 1—figure supplement 1D). We also find a small decrease in Myc protein in the *Imp* RNAi NBs, which is again expected because Imp levels are already very low in *wild type* type I NBs at wL3, and therefore *Imp* knockdown would not have a large additional effect. These results support our model that Imp upregulates Myc protein, but the changes are small due to the small changes in Imp level with the OE and RNAi constructs at the wL3 stage.

Figure 4 – The effect of Imp RNAi, Syp RNAi on myc/transcription is not consistent with this double RNAi being the same as knocking down Imp. It suggests that Syp RNAi has an effect independent of Imp. The authors’ conclusion that the result of this double KD shows "that Imp rather than Syp is the primary regulator of the number of myc^long^." is therefore not supported by the data and needs to be toned down. For this reason, it would also be useful to have ImpOE. Add stage these analyses were done to the legend.

As discussed more extensively above, Imp overexpression with the UAS Imp construct only has a small effect on Imp protein level, compared to the much greater upregulation with the *Syp* RNAi. To address this comment, we have added *myc* smFISH in the Imp OE to Figure 4—figure supplement 1B, but we don’t think it is useful to do extensive quantitations with the Imp OE. We have also added details of the developmental stage to each figure legend.

Figure 4H – "we find that Imp's main direct role is to promote myc^long^ mRNA stability and this results in upregulation of Myc protein, which promotes NB growth and division" however myc half-life is not reduced in Imp RNAi, nor in Imp RNAi_Syp RNAi, but NB size and division rate is reduced in both genotypes (Figure 1A'B'). The authors need to reformulate their conclusion and discuss these inconsistencies.

The reviewer is asking why we see a phenotype in the *Imp* RNAi, but no significant change in *myc* mRNA stability compared to *wild type*. At the wL3 stage, Imp is very low in the *wild type* type I NBs, and therefore knocking down Imp would not be expected to have a very large effect. In the *Imp* RNAi we observe a small but significant decrease in Myc protein (added to Figure 3C) as well as a small but not significant reduction in *myc* mRNA half-life to 0.6-fold of *wild type myc* half-life. This small reduction in Myc expression in the *Imp* RNAi NBs may be sufficient to explain the phenotype of smaller, slower dividing NBs. There may also be an additional effect of Imp-depletion throughout larval development i.e. reduced Imp level leading to less NB growth in early larval stages might be expected to result in smaller NBs at wL3.

Figure 5In Figure 5F authors show Imp levels and myc half-life. The authors claim in several sections, including in Title and Abstract, that "Imp/IGF2BP levels modulate individual neural stem cell growth…" however the authors do not show this in individual type 1 nbs or show any causality or even correlation. Just by looking at 5F there seems to be no correlation between Imp levels and nb area. Comparison between type 1 central brain and MB is interesting, but these are different cell types, and as shown previously, regulated differently. Therefore these 2 cell types combined cannot be used to conclude what the authors claim. The authors need to (a) remove individual cell claims, (b) perform a correlation analysis of myc half-life/IMP levels/nb size in only type 1 nbs, the focus of this work.

The reviewer is correct that in Figure 5 we do not carry out correlation analysis between individual type I NBs and mushroom body NBs at wL3. This point is addressed more fully above, but briefly, we do not think it would be useful to do correlation analysis on type I NBs at wL3 because Imp level is very low in all type I NBs at this stage. Instead, our conclusions on individual NBs are drawn from Figure 6 and Figure 6—figure supplement, using measurements of individual type I NBs at 72 hour ALH. At 72 hour ALH, there is heterogeneous Imp expression, which allows us to examine the effect on different levels of Imp on *myc* mRNA stability and NB behaviour.

We use Figure 5 to draw conclusions between the differences between type I NBs (low Imp) and mushroom body NBs (high Imp). To highlight this difference, we have adjusted the graph in Figure 5F to distinguish the NB types (type I NB with a diamond, MB NB with a circle) in order to clearly emphasise the differences between the NB types.

Figure 5H -Even when type 1 NBs are overexpressing Myc (Figure 3E) they do not reach the average number of progeny labeled as for MB (mycOE type1~4 progeny cells labeled in 4 hours; MB NBs ~6 progeny cells labeled in 4 hours). Therefore, it does not seem that Myc levels are the main cause of this difference. The authors need to rephrase their conclusions when comparing type1 and MB neuroblasts, since these two nb types seem to have many other differences.

This is an interesting observation, but we aren’t confident that our Myc OE is directly equivalent to the high Myc levels in MB NBs. Furthermore, the changing levels of Myc throughout larval development may have some continued effect at the wL3 stage. Nevertheless, to address this comment we have added to our discussion other differences between MB NBs and type I NBs in subsection “Regulated Imp levels control *myc* mRNA 447 stability in individual NBs and NB types".

Include in figure legend (check all legends) what is the exact stage analysed.

Added.

Figure 6In Figure 6 the authors try to address how the differential expression of Imp/Syp throughout time affect myc levels and neuroblasts. They find that at 72hour ALH, Imp::GFP levels are high in nbs and in differentiated cells, but at wandering L3 Imp::GFP levels are much reduced, and that this reduction correlates with a decrease in neuroblast area and myc half-life. However, the authors do not show protein levels of Myc (Myc staining) in these 2 stages, this should be included.The authors should also knock down Imp in 72hour ALH, the stage where Imp is high, and therefore normally playing a role, and see how this affects Myc protein levels.

These suggestions are addressed thoroughly above, briefly we have tested different fixing protocols (methanol fixing and 1% or 4% formaldehyde) as well as different Myc antibody concentrations, but are unable to satisfactorily optimise the Myc antibody staining at 72 hour ALH to perform these experiments. Although this would be a useful addition, we don’t think the Myc antibody staining is essential to support our conclusion that Imp stabilises *myc* mRNA, which we measure directly.

Figure 6F – show NB size too in this graph – to see if indeed there is correlation between myc/imp and cell size at the individual nb level.

This suggestion is discussed fully in the main response. Briefly, the correlation matrix for all five variables is found in Figure 6—figure supplement 1. We have chosen not to add NB size into the Figure 6F (now Figure 6G) graph as we want to focus on the main conclusion that Imp stabilises *myc* mRNA in individual NBs. Referring to the specific correlation requested, we find that *myc* transcript number does correlate with NB size, but Imp level does not correlate with NB size. We highlight this fact in the corresponding Results section, and also discuss the reasons in the Discussion section.

Figure 7 – The insulin signaling or brain sparing is not connected with the process that the authors are studying, i.e. normal temporal regulation of neuroblasts. Remove from this figure as it is confusing. The "individual intrinsic control" is also not clear or supported by the data, remove.

This is a good point, we have added a question mark to the systemic extrinsic signals in Figure 7A and added to the text of the figure legend, to clarify that although this is a plausible model, it has not been directly addressed in the type I NBs.

TitleThe authors are interested in understanding how Imp levels modulates individual neural stem cell growth, however the authors study in general type 1 neuroblasts and do not perform experiments to study how the growth of individual neuroblasts is regulated by Imp. There is one experiment (5F) that characterizes myc half-life, Imp levels and NB area in single neuroblasts, but this experiment does not show if there is correlation among type 1 nbs nor does it show if a difference in levels among type 1 nbs is causal of differences in neuroblast size. For these reasons the Title does not fit with what is really shown in the manuscript, which is how mechanistically Imp levels modulate neuroblast (type 1) growth and division through myc mRNA stability. This needs to be changed.

This is not quite correct, we show experiments in both Figure 5 and Figure 6 measuring *myc* half-life, Imp levels and NB area in individual NBs. We make our conclusions on the role of differing levels of Imp in individual NBs from the data shown in Figure 6 and Figure 6—figure supplement 1. At 72 hour ALH we observed heterogenous Imp expression in type I NBs, and we use this system to show that Imp level is positively correlated with *myc* mRNA stability in individual NBs. In Figure 5F, we make individual NB measurements but do not use these for correlation analysis because the levels of Imp in the type I NBs are always very low in each cell at wL3.